# Mapping specificity, cleavage entropy, allosteric changes and substrates of blood proteases in a high-throughput screen

Federico Uliana[1,7], Matej Vizovišek[1,7], Laura Acquasaliente[2], Rodolfo Ciuffa[1], Andrea Fossati[1],
Fabian Frommelt[1], Sandra Goetze[3,4], Bernd Wollscheid[3,4], Matthias Gstaiger[1], Vincenzo De Filippis[2],
Ulrich auf dem Keller[5] & Ruedi Aebersold[1,6 ✉]

Proteases are among the largest protein families and critical regulators of biochemical processes like apoptosis and blood coagulation. Knowledge of proteases has been expanded by the development of proteomic approaches, however, technology for multiplexed screening of proteases within native environments is currently lacking behind. Here we introduce a simple method to profile protease activity based on isolation of protease products from native lysates using a 96FASP filter, their analysis in a mass spectrometer and a custom data analysis pipeline. The method is significantly faster, cheaper, technically less demanding, easy to multiplex and produces accurate protease fingerprints. Using the blood cascade proteases as a case study, we obtain protease substrate profiles that can be used to map specificity, cleavage entropy and allosteric effects and to design protease probes. The data further show that protease substrate predictions enable the selection of potential physiological substrates for targeted validation in biochemical assays.

---

[1] Department of Biology, Institute of Molecular Systems Biology, ETH Zürich, Zürich, Switzerland. [2] Department of Pharmaceutical and Pharmacological Sciences, Laboratory of Protein Chemistry and Molecular Hematology, University of Padua, Padua, Italy. [3] Department of Health Sciences and Technology, Institute of Translational Medicine, ETH Zürich, Zürich, Switzerland. [4] Swiss Institute of Bioinformatics, Lausanne, Switzerland. [5] Department of Biotechnology and Biomedicine, Technical University of Denmark, Lyngby, Denmark. [6] Faculty of Science, University of Zürich, Zürich, Switzerland. [7] These authors contributed equally: Federico Uliana, Matej Vizovišek. ✉email: aebersold@imsb.biol.ethz.ch

Proteolytic cleavage by proteases is a common protein posttranslational modification and a mechanism that regulates protein functions. It is crucial for cellular health and homeostasis and is also involved in the development and progression of various diseases including cancer, inflammation, autoimmune, cardiovascular and metabolic disorders[1,2]. Therefore, it is not surprising that proteases are widely recognized as diagnostic markers and therapeutic targets in the biomedical field[3]. The knowledge of protease cellular and physiological functions as well as their substrates and cleavage preferences is crucial to design molecules for therapeutic modulation of protease activity[4,5]. In the wake of the progress achieved by bottom-up mass-spectrometry based proteomics[6], several techniques to systematically study protease-substrate relationships have been described. They can be grouped into two broad classes. The first aims at concurrently generating activity profiles of numerous proteases present in a complex sample. This is usually accomplished by the use of activity-based probes[7–9]. The second aims at identifying, typically by mass spectrometry, the substrate(s) of specific proteases, followed by the analysis of protease cleavage products and substrate repertoires, often referred to as protease degradomics[4]. Relevant techniques to identify protease substrates include COFRADIC (combined fractional diagonal chromatography)[10,11], ChaFraDIC (charge-based fractional diagonal chromatography)[12], PICS (proteomic identification of protease cleavage sites)[13,14] and TAILS (terminal amine isotopic labeling of substrates)[15,16] which are reviewed elsewhere[17–19]. More recently, workflows like FPPS (fast profiling of protease specificity)[20] and especially label-free degradomic workflows like DIPPS (direct in-gel profiling of protease specificity)[21] and ChaFraTip (ChaFraDIC performed in a pipet tip format)[22] made protease characterization easier and more accessible by describing simplified workflows and omitting extensive fractionation or labeling steps. Furthermore, DIPPS[21] and ChaFraTip[22] can simultaneously map prime and non-prime substrate sites by sequencing the protease-generated peptides to retrieve protease specificity. In spite of these developments, several limitations remain, which limit the throughput, cost-effectiveness or physiological relevance of these assays. Specifically, the above-mentioned methods suffer from the following limitations: (i) they mostly assay protease-substrate relationships under less- or non-physiological conditions, e.g. using digested (PICS) or denatured proteins (DIPPS) as substrates; (ii) they require chemical modifications, enrichment or separation steps of the protease products (TAILS, COFRADIC, FPPS, PICS, ChaFraDIC), often resulting in costly, time-consuming and technically demanding protocols; (iii) they are not easily multiplexed and usually limited to capturing a few hundred protease cleavages per experiment/sample/fraction which is often not sufficient to comprehensively cover protease-substrate relationships. The consequences of these limitations are well-reflected in the substrates deposited in the MEROPS protease database, currently the most comprehensive protease-substrate resource[23]. About 60% of the 4,000 proteases in MEROPS do not have known substrates (orphan proteases), and less than 200 have more than 30 substrates/cleavages identified to date (Fig. 1a) (MEROPS release 12.1). Since coverage of at least 30 substrates/cleavage events is required to calculate a reliable substrate specificity with an error rate of 5%[24] such analyses are currently only possible for less than 5% of proteases across the kingdoms of life. The expansion of cleavage product datasets generated on proteins with preserved native fold (near-native conditions) will improve our understanding of protease-substrate relationships at two levels. First, the large number of substrates will add statistical power to the calculation of reliable protease recognition sequences and highlight the contribution of substrate steric information to the cleavage pattern. Second, the large number of

substrates will aid the training of algorithms to improve the prediction of proteases involved in natural peptide generation, exemplified by Proteasix[25], PROSPER[26] and SitePrediction[27].

In this study, we report a streamlined method for high-throughput parallel protease characterization, which we dub "High-Throughput Protease Screen" (HTPS). HTPS is based on simple isolation of protease-specific peptides from native lysates using a 96 FASP (96 wells filter-aided sample preparation)[28,29] that are subsequently identified by data dependent acquisition (DDA) mass spectrometry enabling a simultaneous profiling of up to 32 proteases in triplicates. We use HTPS to characterize proteases commonly applied in proteomic workflows (Trypsin, Lys-C, Asp-N, Glu-C and Chymotrypsin), as well as WN NS3, MMP2 and MMP3 as a benchmark. We further identify products of nine blood-activated coagulation proteases (activated α-, β-, and γ-Thrombin, aFVII, aFIX, aFX, aFXI, activated protein C (aPC) and plasmin (PLG); for gene name conversion see Supplementary Data 1), expanding the repertoire of known substrates/cleavage events by about two orders of magnitude (Fig. 1b, Supplementary Data 2) and map the allosteric effect of Na$^+$ on activity, substrate specificity and cleavage entropy. We finally use HTPS data to design fluorescent substrates for activated α-Thrombin and aFX and develop a statistical framework to predict potential physiological substrate candidates of the blood cascade proteases among secreted proteins.

Hence, we describe a simple, high-throughput method for protease product profiling that supports data-driven reconstruction of protease recognition sequences, substrate design, prediction of protease substrates and an assessment of the effects of allosteric changes on substrate specificity.

## Results

**A method for high-throughput screening of protease substrates and cleavage sites on native proteins**. The high-throughput protease screen (HTPS) protocol consists of two main steps: (i) sample preparation and data acquisition, (ii) computational identification and analysis of protease cleavage products (Fig. 1c). First, a native cell lysate is prepared, where endogenous proteases are blocked with low-molecular weight inhibitors and the excess inhibitors as well as peptides resulting from background proteolysis are removed using membrane filters with a 10 kDa MWCO. To screen the protease of interest under microscale conditions, 50 μg aliquots of the thus prepared native lysate are proteolyzed with the protease in question at 1:50 enzyme to substrate ratio. This step is performed in 96FASP filter plates with a MWCO of 10 kDa[29], which retains undigested proteins and the added protease and supports recovery of the cleavage products in the flow-through. Four downstream sample processing steps typical for bottom-up proteomics, namely reduction and alkylation, Trypsin digestion and C18 cleanup are bypassed. The procedure preserves native substrate fold and disulfide bridges as these can impact substrate accessibility while performing proteolysis on proteins in their native fold. The generated samples are free from detergent and salt and the peptides collected after FASP centrifugation are directly analyzed by DDA-MS. This simplifies the sample preparation and the workflow eliminates steps that can lead to peptide loss, induce bias towards a particular class of peptides or alter the protease fingerprints. While trypsinization or an additional digestion step with a complementary protease could be potentially beneficial in a double step proteolysis, we found that the investigated proteases generated a considerable number of peptides and only a generally minor, although variable, amount of under-digested peptides/protein fragments was detected at the end of the reverse phase chromatograms. This does not generally have an impact on chromatography or MS instrument performance, as assessed by

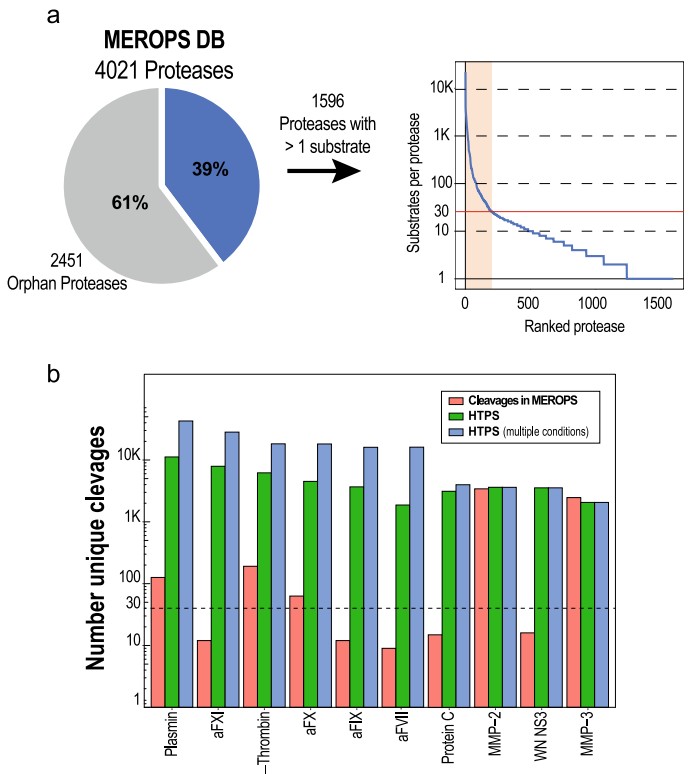

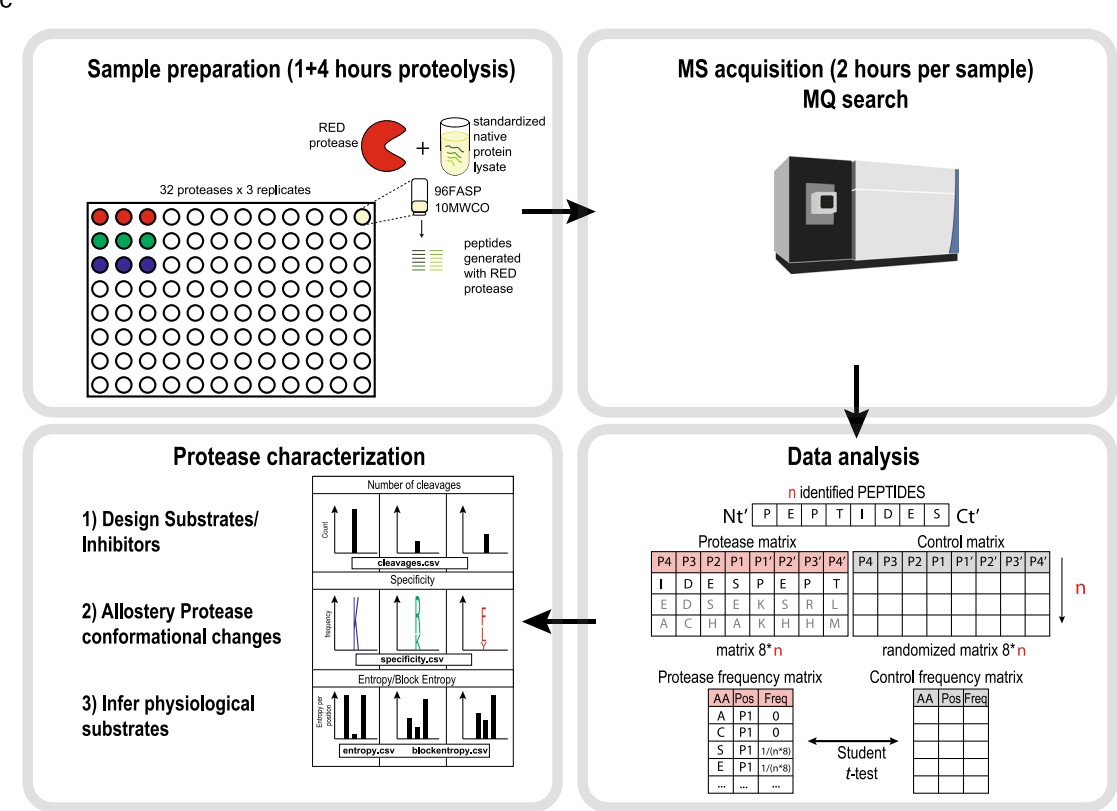

the stability of the retention time and stability of MS2 intensity of the external standard iRT peptides that were used to check the performance of the instrument after every triplicate measurement (Supplementary Fig. 1a, b). The result of these steps are sets of fragment ion spectra of peptides that are highly enriched for substrates of the protease tested.

In the second step, the protease-generated peptides can be identified with any tandem mass spectra search tool. In our implementation the Andromeda[30] search engine in MaxQuant[31] was used with unspecific database search parameters as described elsewhere[21]. Importantly, we searched the data with a reduced database (HTPS_DB.fasta) generated from proteins identified in

**Fig. 1 The protease characterization challenge and the HTPS workflow. a** Distribution of identified substrates per protease annotated in the MEROPS database (release 12.1). From 4,021 proteases reported across the kingdom of life, 2,451 are orphan proteases (protease without known substrates) and only around 200 proteases have a sufficient number of known cleavages (i.e., 30 or more) to calculate their specificity with an error rate of 5% (red line on the chart). **b** Bar plot showing the number of reported protease substrates/cleavages annotated in MEROPS (red) and the number of potential protease substrates/cleavages identified in this study in single ($n = 3$ independent samples, green) or in multiple conditions ($n = 3$ independent samples per condition, blue). **c** High-throughput native microscale protease screen (HTPS). In the screen, a standardized native cell lysate is proteolyzed with the studied protease. The protease-generated peptides are collected, analyzed by MS, and the identified substrate peptides are analyzed to retrieve activity, specificity and cleavage entropy data. This data can be used to (i) design synthetic substrates, (ii) characterize allosteric conformational changes and (iii) infer physiological protease substrates. Source data are provided as a Source Data file.

Trypsin, Lys-C, Asp-N, Glu-C and Chymotrypsin samples. The rationale for this strategy is that searches against large databases with many proteins that are not present/detected in the samples lead to a FDR (false discovery rate) inflation and a decrease in the PSM (peptide-spectrum matches)[32,33], particularly pronounced in case of unspecific searches. In our benchmark, we used HTPS_DB.fasta containing 2,557 protein sequences corresponding to ~12% of the human UniProt database showing the same distribution of amino acids (Supplementary Fig. 2a, b) as the whole UniProt database. With a FDR control at the peptide level set to 0.01, the use of HTPS_DB.fasta increased the number of PSMs compared to a full proteome database by 19% in the case of Trypsin and by more than 33% in the case of Chymotrypsin (Supplementary Fig. 2c). This increased the ratio of matched MS/MS spectra over all MS/MS spectra and the number of identified peptides (Supplementary Fig. 2d), while decreasing analysis time (Supplementary Data 3). Next, positional frequency of amino acids, the cleavage entropy[34] (a quantitative measure of protease specificity) and block entropy[35] (a measure of protease sub-site cooperativity) were calculated with a series of scripts that we developed for the study and that were extensively annotated and deposited in GitHub (https://github.com/anfoss/HTPS_workflow, https://doi.org/10.5281/zenodo.4484341). Protease specificity is a direct result of analyzing the peptide pool generated in a cleavage assay and mapping the determined termini onto the protein sequence. In contrast, cleavage entropy is calculated as information entropy (Shannon entropy) and ranks proteases from less specific (i.e., higher cleavage entropy, e.g., Chymotrypsin in P1) to more specific proteases (i.e., lower cleavage entropy, e.g., Trypsin specificity in P1). These data can be supplemented by block entropy analysis of sequential amino acid blocks upstream/downstream the cleavage site to investigate potential sub-site cooperativity. Briefly, after filtering the MaxQuant peptide results (contaminants, decoys and low-score peptides), the cleavage sequences of the identified peptides were converted to a frequency matrix covering 8 amino acids upstream and 8 downstream the cleavage site (P8-P1 and P1'-P8', respectively). Protease cleavage specificity was inferred by comparing the observed frequency with a random (null) distribution generated from the database and computing a two-side paired t-test. The cleavage sequences from peptides identified after proteolysis were directly used as input for the protease characterization, as we did not introduce a bias from the original protein termini (Supplementary Fig. 3a–d). Furthermore, we observed that the protease specificity profile was not influenced by background peptides present in absence of a protease (Supplementary Fig. 4a) because they were present in low numbers and no significant positional enrichment of amino acid frequencies compared to a random (null distribution) was observed (Supplementary Fig. 4b). Importantly, the number of peptides identified by HTPS provides a good proxy for monitoring protease activity as demonstrated by the global proteolysis kinetics of Chymotrypsin (Supplementary Fig. 5a) and α-Thrombin (Supplementary Fig. 5c). Of note, while the number of detected cleavage products increased over time, the specificity

inferred from the detected cleavage products was mostly time-independent for Chymotrypsin (Supplementary Fig. 5b) and α-Thrombin (Supplementary Fig. 5d). Overall, the combined experimental and data analysis HTPS workflow supports, in a single operation, the identification of thousands of cleavage events, outperforming for almost all proteases the number of reported substrates/cleavages in MEROPS database. This is particularly noteworthy in the case of aFXI, aPC and aFVII proteases for which, so far, less than 30 substrates/cleavages were identified (Fig. 1b, Supplementary Data 2).

**Benchmarking the performance of the HTPS screen.** To test the performance of HTPS we conducted three distinct benchmarking experiments. First, we applied our protocol to the proteases Trypsin, Lys-C, Asp-N, Glu-C and Chymotrypsin which are specific, well characterized[36] and commonly used in proteomic workflows. Protease characterization was performed in triplicates using the test proteases at a [E]/[S] ratio of 1:50 and the lysates as substrate sample. Using the workflow described above we identified a higher number of cleavage events for proteases specific for basic amino acids compared to proteases with other cleavage specificities: we identified around 16,600 and 14,000 peptides with Trypsin and Lys-C, respectively, with an overlap of 91.6% and 86.8% between the triplicates (Fig. 2a). For proteases recognizing amino acids with acidic side chains (Glu-C and Asp-N) and for proteases with lower specificity like Chymotrypsin, we recovered between 8,800 and 9,400 peptides, with similar levels of reproducibility (average overlap of 88%) between triplicates (Fig. 2a). Non-tryptic peptides often have worse chromatographic separation, ionization and fragmentation properties than Trypsin products and it is estimated that only 4% of all proteomic data sets are generated with proteases other than Trypsin[37]. Nevertheless, the fraction of matched MS/MS spectra over all MS/MS spectra range between 14–27% for all analyzed proteases (Supplementary Data 3). From the list of identified cleavages we generated specificity profiles via iceLogos[38] with a p-value cutoff of 0.01 (Fig. 2b, Supplementary Data 4 and 5, for heat maps see Supplementary Fig. 7) in agreement with data from the MEROPS database[23] and in-line with their well characterized cleavage specificity profiles[36].

Second, to further benchmark the performance of HTPS against established methods, we used it to characterize the substrate specificity of a viral protease. WN NS3 is a serine protease from a pathogenic West Nile flavivirus that mostly causes flu-like symptoms. While viral proteases are promising therapeutic targets, their characterization is difficult due to lack of structural information and a rather high degree of substrate specificity (they usually process a large viral polyprotein)[39] and to date only 16 substrates are reported in MEROPS. Recently, the WN NS3 protease specificity was extensively characterized by the use of fluorescent combinatorial libraries, reporting a strong preference for basic residues like Arg and Lys at P1, and a preference for Lys at P2 and P3 position[40]. While this study included more than 100 natural

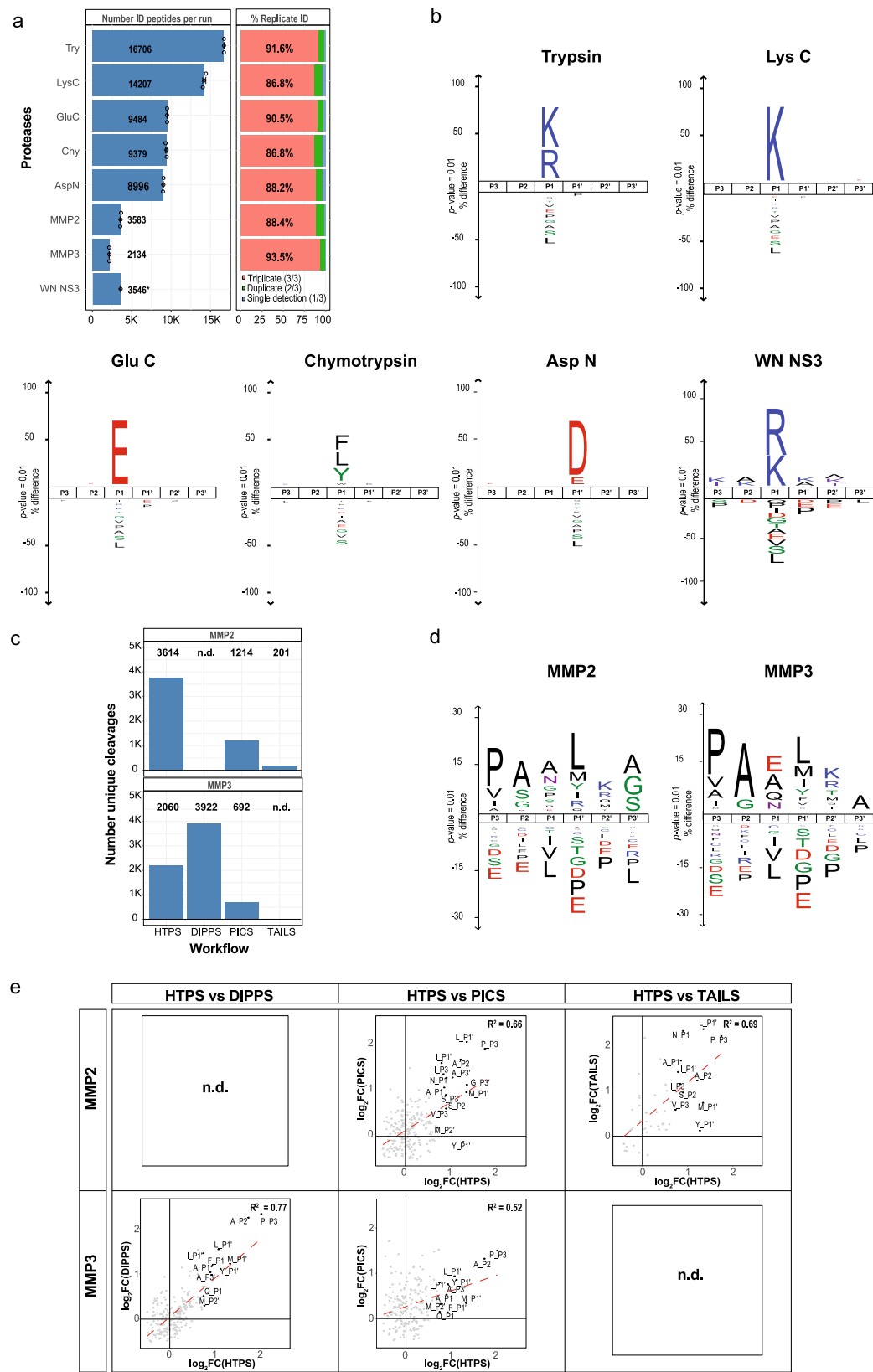

and unnatural amino acids in the combinatorial libraries, the positional preferences identified with our screen for the natural amino acids are in good agreement, revealing a trypsin-like specificity of the WN NS3 protease (Fig. 2b).

As the third benchmarking step we characterized MMP2 and MMP3 from MMPs family which have been extensively studied by multiple proteomics techniques[41] because of their involvement in development and progression of different pathologies, especially cancer[42]. Over the last few years, the substrate specificity of MMPs was characterized using different methods, including PICS[43,44], TAILS[45] and DIPPS[21]. For the comparison, we used the cleavages identified in the studies and analyzed them

**Fig. 2 Benchmark of HTPS performance with different proteases. a** Peptides generated from benchmark measurements using well-characterized proteases, WN NS3 protease and MMPs; each protease is characterized by the average number of peptides identified from three independent replicate experiments (left) and by the overlap across triplicates (right). For all proteases except WN NS3 ($n = 1$) three independent replicates were analyzed. Data are presented as mean values±SD. **b** Specificity benchmark with proteases commonly used in proteomics workflows as well as WN NS3 protease and MMPs presented as iceLogos. The protease specificity preferences are shown for P3-P3' positions. **c** Identified cleavages for MMP2 and MMP3 using different protease characterization approaches (PICS[43,44], TAILS[45], DIPPS[21] and HTPS). HTPS analysis was performed with three independent replicates. **d** MMP2, MMP3 substrate specificity presented as iceLogos covering P3-P3' positions. **e** Correlation of the reported specificity enrichment per position for MMP2 and MMP3 between HTPS and other protease workflows (DIPPS[21], PICS[43,44] and TAILS[45]). Source data are provided as a Source Data file.

with the HTPS workflow to generate the frequency matrices as the basis for the respective positional specificities. In three out of four comparisons, our approach resulted in a higher number of cleavages (Fig. 2c), 3,614 for MMP2 and 2,464 for MMP3 and an overlap of almost 90% between triplicates (Fig. 2a, Supplementary Data 4). We determined all positional amino acid (AA) enrichments in comparison to the natural AA distribution in the database, reporting only significant values (adjusted $p$-value <0.01, Fig. 2d, Supplementary Data 5). HTPS data indicated that MMP2 and MMP3 both have similar specificities (Fig. 2d), with preference for Pro, Ala, Val and Ile at P3 position. Further, at P2 position, we observed a preference for Ala, Ser, Gly for MMP2 and Ala, Gly for MMP3. At P1 position Ala, Asn and Pro was observed for MMP2 and Glu, Ala, Gln and Asn for MMP3. Additionally, we observed a preference for Leu, Met, Tyr, Ile at P1' and Lys, Arg, Met, Thr at P2' for both proteases and a different specificity in position P3' for Ala (in case of MMP3) and for Ala, Gly and Ser (in case of MMP2), which is mostly in agreement with the proteases preference reported in other aforementioned studies[21,43–45]. The comparison of the methods in terms of reported positional amino acid enrichments showed a good overall correlation between PICS, TAILS, DIPPS and HTPS with the $R^2$ ranging from 0.52 for PICS to 0.77 for DIPPS (Fig. 2e). Taken together, these observations corroborate the validity of HTPS as an alternative method for protease profiling.

**High-throughput screening of blood coagulation cascade proteases.** We then applied the method to comprehensively characterize the blood cascade serine proteases. The group of enzymes tested consists of blood coagulation proteases aFVII, aFIX, aFX, aFXI, activated α-Thrombin, PLG and aPC as well as β- and γ-Thrombin. We chose these proteases because they (i) are biologically and chemically related; (ii) have a substantial therapeutic potential; (iii) have been to some extent structurally characterized and (iv) their repertoire of substrates is not yet fully characterized. Moreover, we extended the screening to β- and γ-Thrombin, two proteolytic proteoforms of α-Thrombin. Albeit not physiologically relevant in the coagulation cascade, they are a good example to test the sensitivity of HTPS with protease pro-teoforms. In blood cascade, the concerted action of serine proteases regulates blood clot formation through activation of Thrombin which converts fibrinogen to insoluble fibrin and activates platelets via PAR1 proteolytic activation[46]. Besides the nine blood coagulation proteases we also included Chymotrypsin to the screen because it has the archetypal protease structure for the S1 chymotrypsin-like family[47].

The respective proteases were analyzed using the HTPS workflow. The activities of all coagulation proteases included in the screen were determined by active site titration (Supplementary Data 1) in order to standardize the activity of proteases used in the assay. The detected specificity features are summarized in Fig. 3a. For each protease we identified from 1,800 for aFVII and up to more than 10,000 peptides for PLG (Fig. 3b). This represents an increase in the number of identified cleavages by about two orders of magnitude compared to MEROPS database (Figs. 1c and 3b, Supplementary Data 2). While most of these substrates are not likely to be processed during blood coagulation

due to the nature of the substrate sample, they are nevertheless very useful to determine the cleavage specificity, cleavage entropy, allostery and other functional/structural properties of the proteases. The heat maps shown in Fig. 3a and Supplementary Fig. 7 report significant (corrected $p$-value < 0.01, Supplementary Data 5) enrichment of amino acids around the cleavage site for positions P4-P4', in comparison to the amino acid distribution in HTPS_DB.fasta. To gain a more structured insight into the cleavage specificity relationships among the tested proteases, we performed an unsupervised hierarchical clustering according to their substrate preferences (Fig. 3c). This analysis revealed the existence of 4 clusters. Cluster 1 included PLG, aFXI and γ-Thrombin, proteases with a strict specificity limited to position P1 for Arg and Lys. Cluster 2 included α- and β-Thrombin, and aFVII and showed specificity in P1 for Arg and a contribution to the specificity of all positions close to the cleavage site (P3-P2'). Cluster 3 contained aFX, aPC and aFIX and showed an intermediate specificity between the first two clusters, but generally closer to cluster 1. As expected, Chymotrypsin clustered separately from the clotting proteases as it shows a cleavage specificity for hydrophobic amino acids (Phe, Trp and Tyr at P1 position and Leu, Met to a lesser extent).

Both, specificity profiles and clustering are in-line with the prior knowledge about these proteases from MEROPS and other specificity studies[48,49]. All coagulation proteases have a defined trypsin-like specificity in position P1. There is also a strong preference for substrates with Arg and to a lesser extent for Lys in P1 position where specificity is tightly regulated by the ionic interaction between the negative carboxylate group of Asp 189 and the positive charged group of the substrate[50,51]. While the detected enrichment of Arg at P1 position was similar for all proteases, the level of enrichment of Lys was highest for the members of cluster 1. All profiles are characterized by higher specificity in P1 position and lower specificity in other extended positions, indicating that, similar to Trypsin and Chymotrypsin, the protease specificity is determined mainly by the P1 position. This is also evident from the sub-pocket resolved cleavage entropy profiles (Fig. 3d), which show the substrate preference per position for each protease. Proteases from cluster 1 are promiscuous proteases, their specificity is essentially determined by the amino acid in position P1 and, as consequence, they cleave more frequently compared to other coagulation proteases (Fig. 3b).

Coagulation proteases differ from Chymotrypsin structurally by the presence of two insertion loops (loop 60 and loop 148)[46]. These loops form a rigid lid-like structure which regulates accessibility to the catalytic pocket, generates a more extended specificity for coagulation proteases (from P3 to P2') and thus a distinct substrate fingerprint. As an example, the preference of aFX for Gly in position P2 (Fig. 3a) is generated by bulky residues in the insertion loop which accepts small amino acids at the corresponding substrate positions[52]. In α-Thrombin, the 60-loop generates the preference in position P2 for Pro, hydrophobic and planar residues and a preference in position P1' for small residues like Ala (Fig. 3a). The key role of the steric hindrance of the 60-loop in the selectivity

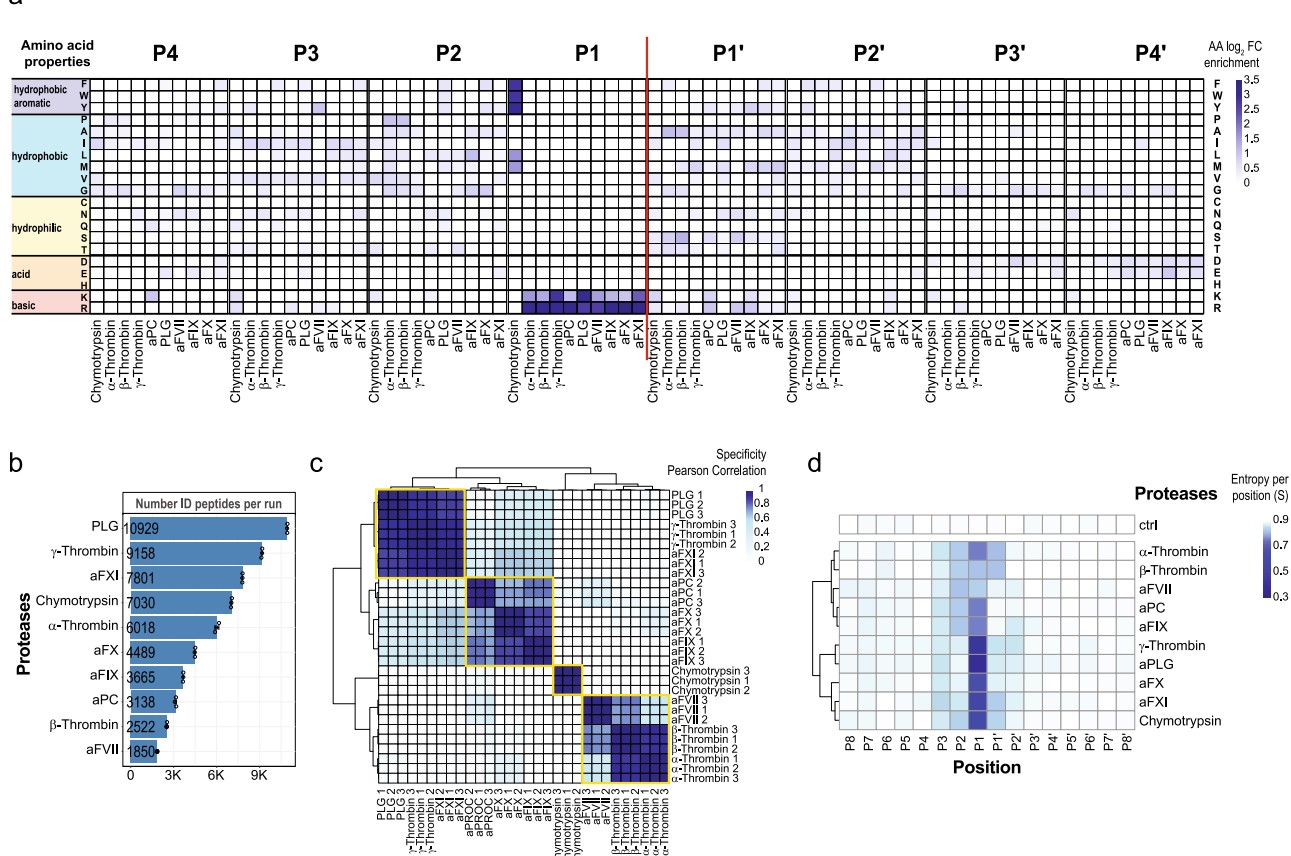

**Fig. 3 High-throughput screening of coagulation proteases. a** Positional substrate preferences of coagulation proteases from the chymotrypsin-like family (Chymotrypsin, activated α-, β-, γ-Thrombin, aFVII, aFIX, aFX, aFXI, aPC and PLG). The heat map includes positions P4-P4'. The AAs are grouped according to their physicochemical properties. The enrichments are reported as log$_2$ FC compared to a random AA distribution generated from HTPS database. **b** Peptides generated by the coagulation proteases included in the study; each protease is characterized by the average number of peptides identified from three independent replicate experiments ($n = 3$). Data are presented as mean values±SD. **c** Unsupervised hierarchical cluster of coagulation proteases. The color scale describes the Pearson correlation coefficient value calculated for the respective protease samples. **d** Unsupervised hierarchical cluster of coagulation proteases according to the positional cleavage entropy. The color scale describes the cleavage entropy (S) values for positions P8-P8' for all respective proteases included in the assay. Source data are provided as a Source Data file.

of α-Thrombin was shown by mutagenesis experiments[53] and by the promiscuous specificity of γ-Thrombin generated by autoproteolysis of α-Thrombin. These cleavages generate an extensive disorder region in the 60-loop which provides an explanation for the observed loss of specificity[54].

So far, different techniques have been applied for in-depth characterization of the substrate specificity of α-Thrombin, including combinatorial peptide libraries[55–57] and phage display libraries[58]. In this study, we accurately recapitulated the well characterized α-Thrombin specificity and compared it in the so far most comprehensive fashion with the other blood coagulation proteases included in the study. Notably, we increased the knowledge about their substrates/cleavages by a large margin (Supplementary Data 2 and 4), defined their specificities (Supplementary Fig. 7, Supplementary Data 5), cleavage entropies (Supplementary Fig. 8, Supplementary Data 6), and block cleavage entropies (Supplementary Figs. 9 and 10, Supplementary Data 7) to show that this unique dataset could recapitulate and extend the knowledge on blood proteases.

**Detection of effects of modulators on activity and specificity of blood cascade proteases**. In the previous section, we found that

specificity profiles generated with our method were sensitive enough to detect subtle differences between the investigated proteases. We next asked whether HTPS could detect changes of activity and specificity profiles after the binding of modulators, which can influence the catalytic activity of an enzyme (cofactor) or induce allosteric rearrangements of the protease catalytic pocket. We investigated the effect of Tissue Factor, a cofactor that binds to FVII to form a protein complex that activates the protease and thus initiates the extrinsic pathway of coagulation[59]. We observed that the presence of Tissue Factor boosted the activity of aFVII resulting in a 3-fold increase in FVII generated peptides (from an average of 2,814 to 6,912) (Fig. 4a). Next, we applied HTPS to study the effects of Na$^+$ binding on the activity of coagulation proteases and to investigate protease specificity changes as consequence of the allosteric mechanism. Allostery is a crucial regulatory mechanism of proteins where the binding of an allosteric effector modulates conformational and consequently functional changes of a protein. Allosteric effects have been extensively studied in the case of α-Thrombin[60,61], but a systematic proteomic approach has not been applied to study the effect of Na$^+$ on coagulation proteases. To investigate the effects of Na$^+$ on the reorganization of the hydrophobic pocket in the active site of blood coagulation proteases and the ensuing effects on activity and specificity, we generated protease fingerprints of

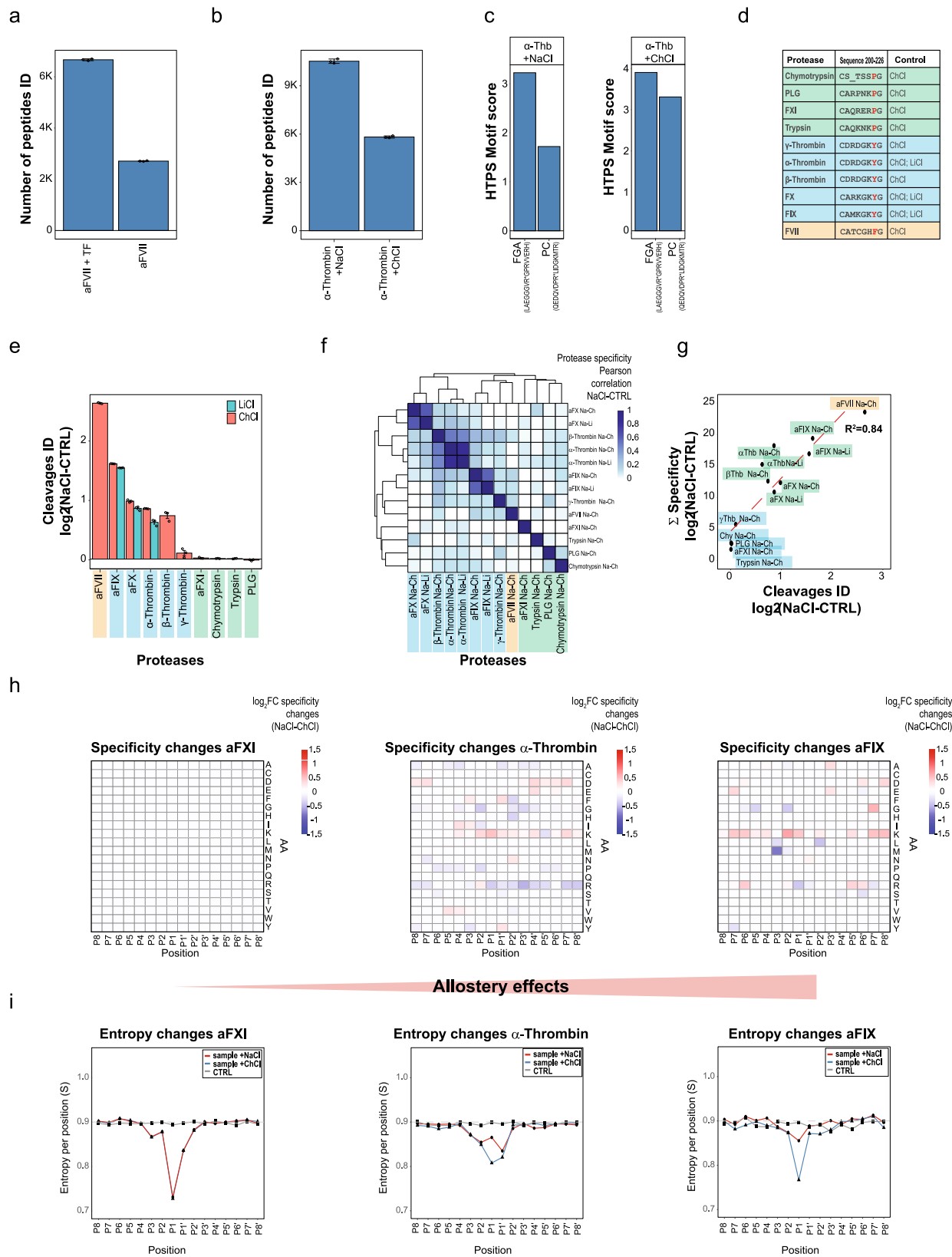

tested coagulation proteases in the presence of 0.2 M NaCl and choline chloride (ChCl). As the latter was reported to be a weak competitive inhibitor of protease activity of aFX[62], we measured the allosteric effect of Na$^+$ using double controls with 0.2 M ChCl or LiCl to keep the ionic strength constant without exerting an allosteric effect[63]. While Ch$^+$ is a bulk monovalent cation, which

cannot be coordinated in the Na$^+$ binding site, Li$^+$ is too small to generate a productive allosteric effect. Before assaying the effect of Na$^+$ allostery on protease activity and specificity at 20 °C (at this conditions, α-Thrombin exists predominantly in Na$^+$-bound or Na$^+$-free form[63]), we evaluated the effect of lower temperature on activity and specificity of Chymotrypsin and α-Thrombin. While

**Fig. 4 HTPS detects activity and specificity changes induced by allosteric modulators. a** The effect of Tissue factor on the number of identified peptides with aFVII ($n = 3$ independent replicates). Data are presented as mean values±SD. **b** The effect of Na$^+$ on the number of identified peptides with α-Thrombin ($n = 3$ independent replicates). Data are presented as mean values±SD. **c** The effect of Na$^+$ on amino acid specificity enrichment of α-Thrombin towards its known physiological substrates (FGA and PC). The specificity change was calculated from the sum of significant fold change of amino acid enrichment per position (HTPS Motif Score). This value reflects the preference of α-Thrombin forms towards the respective substrates. **d** Chymotrypsin-like family protease primary sequence alignment in correspondence of the sites which regulate the coordination of Na$^+$ (220–226), adapted from[67]. Residue 225 is crucial for Na$^+$-induced allosteric regulation of serine proteases. Proteases with Tyr (blue) or Phe (orange) in position 225 can coordinate the Na$^+$ ion while proteases with Pro (green) in position 225 cannot bind it. **e** Relative log$_2$FC of identified cleavages in presence and absence of Na$^+$ ($n = 3$ independent replicates). Data are presented as mean values±SE. The color code corresponds to the allosteric requirements reported in 4d, while the bar colors correspond to the control used (ChCl - red, LiCl - blue). **f** Unsupervised hierarchical cluster of protease specificity changes as result of allosteric effects (NaCl-ChCl/LiCl). The color code corresponds to the allosteric requirements reported in 4d. **g** Correlation plot between the changes of identified cleavages and changes of substrate specificity observed as result of Na$^+$ allosteric interaction. The color code corresponds to the allosteric requirements reported in 4d. **h** Heat maps of specificity changes generated by Na$^+$ allostery: aFXI, activated α-Thrombin and aFIX are ordered based on the magnitude of allosteric effects at the specificity level. While aFXI does not show any changes, aFIX shows significant changes on the level of specificity. **i** Positional entropy changes generated by Na$^+$ allostery: aFXI, activated α-Thrombin and aFIX are ordered based on the magnitude of entropy changes (S) observed at the positional entropy level. Source data are provided as a Source Data file.

we observed around 20% reduced number of cleavage products (Supplementary Fig. 6a, c) we did not detect a significant change in protease specificity (Supplementary Fig. 6b, d), thus confirming that HTPS can be efficiently applied to study coagulation proteases under optimal allostery conditions. We first tested whether the experimental conditions recapitulated the well-known activity patterns of α-Thrombin towards Fibrinogen (FGA) and Protein C (PC). Previous studies have shown that α-Thrombin, when bound to Na$^+$ (Fast form) has an enhanced activity towards the proteolysis of fibrinogen to fibrin, crucial for clot formation (pro coagulant activity). When α-Thrombin is in the Na$^+$-free form, and co-adjuvated by Thrombomodulin, it can cleave and activate PC, a protease that influences α-Thrombin generation via a negative feedback mechanism[64–66] (anticoagulant activity). This equilibrium is particularly relevant at 37 °C, as the dissociation constant of α-Thrombin with bound Na$^+$ is close to the concentration of the ion in blood[64] and a subtle deviation of Na$^+$ concentration, e.g., around platelet thrombi in vivo, generates a different substrate selectivity with an important implication for the pro- vs. anti-coagulant activity of α-Thrombin. The number of identified peptides confirmed a higher activity for α-Thrombin in the Na$^+$-bound form (Fast form) compared to the Na$^+$-free form (Slow form) (Fig. 4b). The cleavage patterns observed in our data were used to calculate the sum of significant positional enrichment of each AA against a random distribution (HTPS Motif Score) for physiological substrate motifs of FGA (LAEGGGVR-GPRVVERH) and PC (QEDQVDPR-LIDGKMTR)[58]. In presence of NaCl, we observed a clear preference for FGA in comparison to PC substrate; while in presence of ChCl we did not observe such a preference (Fig. 4c). This is in agreement with previous studies, which showed that 0.2 M Na$^+$ led to an increase in the specificity towards FGA, but not towards PC[64]. This effect, together with boosted activity of α-Thrombin in the presence of Na$^+$ (Fig. 4b) results also in a higher rate of FGA cleavages.

Other proteases, similar to α-Thrombin, exhibited an increase in proteolytic activity and these patterns reflected well the requirements for allosteric regulation of blood cascade proteases, where Na$^+$ can be coordinated only if Phe or Tyr are at position 225[62,67,68] (Fig. 4d). Indeed, we observed the strongest fold change of activity (Fig. 4e) between NaCl and ChCl/LiCl in case of aFVII (Phe at position 225), followed by aFIX, aFX, activated β-, α- and γ-Thrombin (all having Tyr at position 225). In contrast, for Trypsin, aFXI, Chymotrypsin and PLG, proteases with a Pro in position 225, we observed no significant differences between NaCl and ChCl (Fig. 4e). Interestingly, we observed that γ-Thrombin, which contains a Tyr in position 225 showed a

somewhat intermediate pattern between the two groups, presumably due to the flexibility of the 60-loop which generates a reduced selectivity in the catalytic pocket (Figs. 3a and 4e).

We next performed an unsupervised hierarchical clustering to investigate the impact of allostery on the specificity of blood coagulation proteases included in the study. We clustered the significant specificity changes detected in presence of NaCl vs. ChCl/LiCl (Fig. 4f). We observed that i) the cleavage events detected in ChCl and LiCl controls clustered closely together, indicating the effects observed on protease activity are a direct consequence of Na$^+$ and ii) proteases regulated allosterically by Na$^+$ clustered closely together and showed significant changes in their substrate specificity. In contrast, no significant changes were observed for proteases that cannot bind Na$^+$ and thus cluster separately. The specificity differences observed in case of aFVII, aFIX, aFX, β- and α-Thrombin suggest, that Na$^+$ had an impact not only on the number of cleavages, but also on substrate specificity (Supplementary Figs. 7 and 11). Importantly, this is also evident from the correlation between activity changes and the changes detected at the level of substrate preference with a R$^2$ value of 0.84 (Fig. 4g). These results demonstrate that Na$^+$ binding to the allosteric site of coagulation cascade proteases regulates their activity in a way that reflects on protease substrate preference (Fig. 4h) and cleavage entropy (Fig. 4i) with strong changes observed, for example, for aFIX, moderate changes for activated α-Thrombin and no changes observed for aFXI (for other proteases, see also Supplementary Figs. 7, 8, 9, 10 and 11).

Overall, we characterized the effect of Na$^+$, a critical and well-known allosteric binder and regulator, on coagulation proteases included in the study, and showed that HTPS has the capacity to detect the functional consequences of allosteric changes with remarkable sensitivity. We found that differential activity, specificity and entropy correlated exactly with the presence or absence of residues that enable Na$^+$ coordination. This highlights the potential of this approach as a tool for systematic screening of the effects of drugs or peptide-mimetic molecules as modulators of therapeutically relevant proteases.

**Designing fluorescent substrates for blood cascade proteases.** To demonstrate the translational value of HTPS, we used the protease specificity data generated above to design fluorescent substrates to detect and discriminate the activity α-Thrombin and aFX. The design of fluorescently or otherwise labeled substrates is important to characterize proteases in kinetic cleavage assays and to use this knowledge to support the design of new activity-based probes and inhibitors. After characterizing the specificity of the proteases, we selected for each position amino acids with most

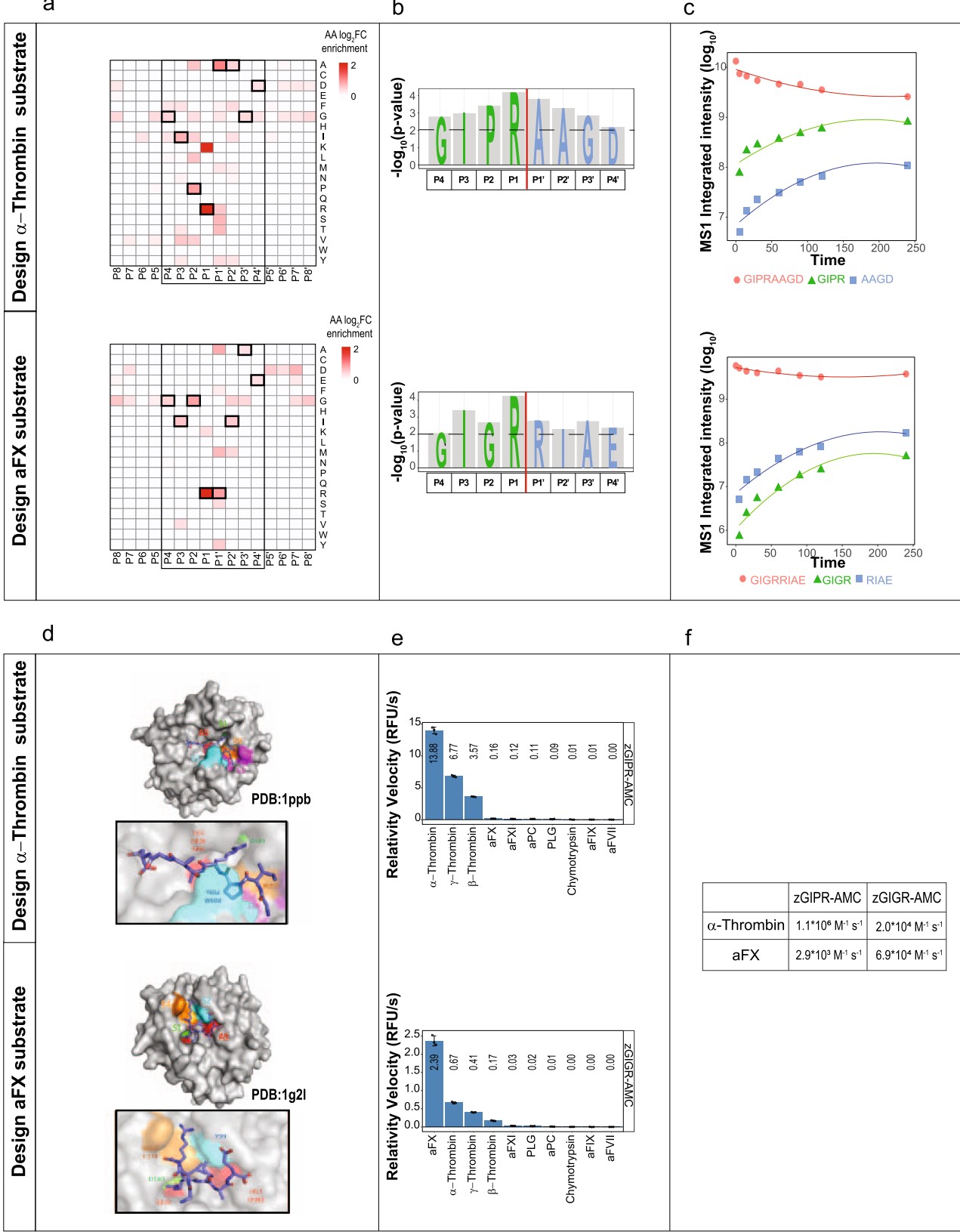

significant positional enrichment (Fig. 5a, b). We designed two synthetic peptides that represent the best match according to the detected specificity for activated α-Thrombin (NH2-GIPR↓AAGD-COOH) and aFX (NH2-GIGR↓RIAE-COOH). As our analysis investigated the positional specificity but did not take into account possible sub-site cooperativity[57], we confirmed that these peptides were cleaved effectively by the respective proteases. We monitored the intensity of the cleavage products by mass spectrometry, using MS1 signal integration (Fig. 5c). The results showed the expected patterns and thus confirmed that both synthetic peptides represent a good entry point for development of substrates.

**Fig. 5 Design of octapeptides and fluorescent substrates. a** Determination of best-matched substrates from the positional amino acid preferences using positional substrate preferences ($\log_2$FC enrichment compared to random distribution). The best-matched positions selected for the substrate design are highlighted in bold squares. **b** Representation of $-\log_{10}$ p-values of AA selected for the octapeptide design to assay protease activity of α-Thrombin and aFX. **c** The MS1 intensity integrated area for targeted octapeptide substrates and the corresponding cleavage fragments after incubation with activated α-Thrombin and aFX over time (0–240 min). **d** In silico docking of activated α-Thrombin and aFX with the two model octapeptides obtained by HPEPDOCK software. The docked peptides are shown in stick mode and proteases (α-Thrombin 1ppb; aFX 1g2l) in surface representation. The location of active site (AS) and specific active pocket sites (S1–S4) are indicated in different colors (green, orange, red and purple). **e** Determination of substrate selectivity measured by the relative reaction velocities (RFU/s) of fluorescent substrate processing for activated α-Thrombin and aFX tested with a panel of closely related coagulation proteases ($n = 3$ independent replicates). Data are presented as mean values±SD. **f** The calculated $k_{cat}/K_M$ values ($M^{-1}s^{-1}$) for activated α Thrombin and aFX. Source data are provided as a Source Data file.

To evaluate the exact mode of binding of these peptides to the protease active site (AS), we performed a molecular docking analysis. The structural data for α-Thrombin (1ppb) and aFX (1g2l) showed strong similarities of the S1 specific pockets (Asp 189), whereas the S2 and S4 subsites were characterized by distinct topologies. While S2 is covered by the 60-insertion loop in α-Thrombin, it is smaller and solvent accessible in aFX. Further, the aryl-binding site S4 of α-Thrombin, located above the conserved residue Trp 215, is lined by residues Leu 99 and Ile 174. In aFX, the S4 subsite is built by the corresponding residues Tyr 99 and Phe 174, which together with the indole ring system of Trp 215 form the walls of an aromatic box. Our docking models (Fig. 5d) showed how the residue P1 (Arg) can be effectively oriented inside the S1 pocket, while the different P2 residues (i.e., Pro in the case of α-Thrombin and Gly in the case of aFX) can fit specifically in the correspondence of S2 subsites, thus ensuring the interaction with the AS. These results indicate that the substrate-design based on HTPS results produces structurally plausible solutions.

Next, we designed small fluorescent tetrapeptide substrates corresponding to the P4-P1 active site preferences to monitor the activity of activated α-Thrombin (zGIPR-AMC) and aFX (zGIGR-AMC). While some selectivity is lost because of placing the fluorophore at P1'-P4' positions, we tested the substrates against a panel of proteases in a standard assay[69] and observed that both substrates had selectivity for the target proteases (Fig. 5e). Accordingly, zGIPR-AMC was most efficiently cleaved by α-Thrombin and also by γ- and β-Thrombin, while other proteases included in the assay did not cleave zGIPR-AMC. The zGIGR-AMC was less selective because it was cleaved by aFX and also by α-, β- and γ-Thrombin. We further calculated the $k_{cat}/K_M$ and demonstrated that zGIPR-AMC had good selectivity for α-Thrombin over aFX (380-fold higher $k_{cat}/K_M$). Selectivity in case of zGIGR-AMC was substantially lower with $k_{cat}/K_M$ 3.5-fold higher values for aFX in comparison with α-Thrombin (Fig. 5f). The measured kinetic parameters of the Thrombin substrate were in the same range as commercially available[70] and other reported substrates[71]. Accordingly, the α-Thrombin substrate had a 1,000-fold higher $k_{cat}/K_M$ values compared to H-β-AGR-pNA and a 180-fold higher $k_{cat}/K_M$ value compared to zGGR-AMC, two commercial substrates used in Thrombin generation assays[70]. The chromogenic Thrombin substrate S2238 (H-(D)-Phe-Pip-Arg-pNA), which has physicochemical properties similar to zGIPR-AMC, is 23-fold more selective than our zGIPR-AMC[72], because it contains in position P3 a non-natural amino acid (D-Phe) and Pip (Pipecolic acid, i.e. homoproline) at P2 position that provide additional selectivity in the hydrophobic pocket. This example demonstrated the translational value of HTPS screen, where it is possible to generate a substrate with reasonable selectivity towards the investigated protease in a simple and straightforward way, and without extensive testing or large peptide libraries.

**Using HTPS data to predict physiological substrates**. For the extensive characterization of the selected target proteases in this study, a full native lysate from HEK293 cells was used. Since coagulation cascade proteases are known to be secreted and their relevant substrates are primarily found in blood, our characterization is likely to have captured a large number of biochemically plausible, but physiologically irrelevant substrates/cleavages. As a final validation step of our protocol, we therefore asked whether we could use the protease specificity information derived from a native lysate to generate hypotheses on proteins known to be secreted into the blood. To pursue this aim, we developed a three-step filtering framework to single out, from a large initial search space, substrate candidates for specific proteases (Fig. 6a). We used as a reference the 718 proteins reported as secreted to the blood (human blood secretome from ProteinAtlas[73,74]) which contain ~0.3 million 8-residue sequence combinations (from P4-P4'). In the first step of the analysis, we used a filtering strategy based on the HTPS-generated protease data. Specifically, we calculated positional enrichment of each AA against a random distribution generated from the database and scored each of the potential target sequences using the sum of the significant fold changes associated with the respective residues (HTPS Motif Score) as shown on Fig. 6b. This step correctly identified 13 well-known Thrombin natural targets (PC, FVIII, IGFBP5, FV, FXI, FGA and FGB)[58,66] (Fig. 6c) in the top 1% of candidate substrates. We observed that the distribution of Motif Score was bimodal for promiscuous proteases with high specificity in P1 position (i.e. lower cleavage entropy at the cleavage site) such as Trypsin and PLG (Supplementary Fig. 12b, c), and much less discrete for those showing less promiscuous specificity features, e.g. α-Thrombin (Fig. 6b) and aFVII (Supplementary Fig. 12a). This indicated that for the latter class, there was not a discrete population of preferred substrates, but broader specificities modulated by the combination of all amino acids (cooperativity effects). Using MEROPS (release 12.1) substrates[23] identified in the secretome as true positives, we constructed receiver-operator curves and filtered the data to match a false positive rate of 1% (Fig. 6d). By this means, we could reduce the search space of potential physiological substrates by about 100 times. The evaluation of the sensitivity and specificity indicated a good prediction power for all proteases included in the analysis (average AUC~0.97, Supplementary Fig. 12d–j). In a second step, we used the JPred4[75] software tool to predict the solvent accessible regions of (nearly) all 718 secretome proteins and removed all the sequences that were predicted to be buried, thus eliminating structurally implausible targets. This step further reduced the number of potential targets by about a half, from 2,695 to 1,385 (in case of α-Thrombin substrates). Finally, in a third step, we used loose protein-level filters to further refine the target selection: proteins for which no expression was measured as well as those for which no co-citation with the target protein was reported were removed. A good proxy for physiological substrates

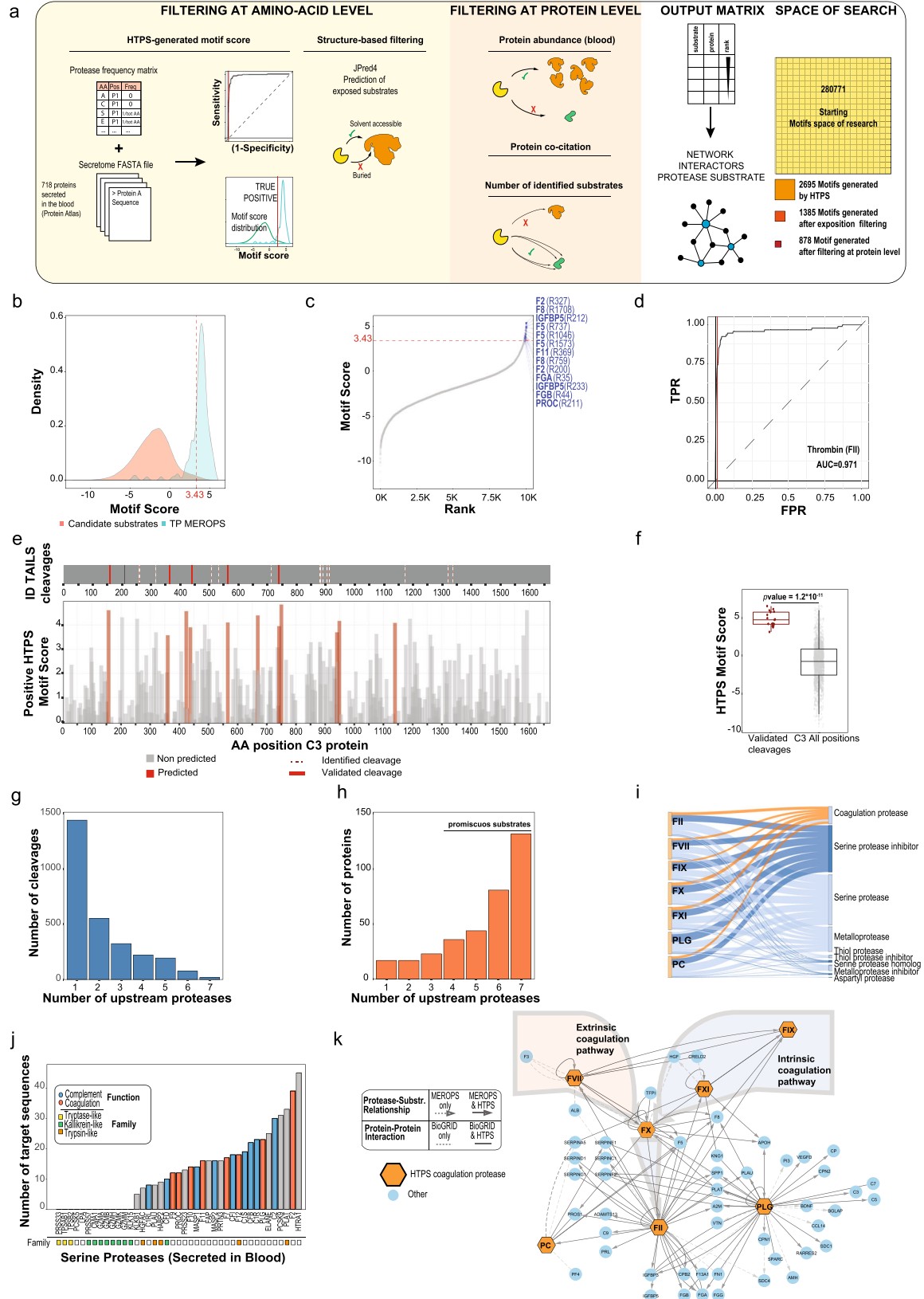

was calculated from the ranking of the frequency of co-citation, protein abundance and the number of potential cleavage sites. These steps significantly reduced the search space (we identified for Thrombin 878 potential physiological cleavages), while having a negligible effect on the recall of previously known substrates (Supplementary Fig. 12k). As expected, the final score generated

from our filtering strategy was highly skewed towards known substrates reported in MEROPS, indicating that it correctly reports potential substrate candidates (Supplementary Fig. 12m, Supplementary Data 8). To demonstrate the predictive power of HTPS, we selected the α-Thrombin predicted cleavage sites on C3 protein. C3 protein, already reported to be α-Thrombin substrate

**Fig. 6 Identification of physiological substrate candidates from HTPS data. a** Three step-filtering framework to identify candidate substrates in the blood secretome. All proteins annotated in ProteomeAtlas as part of the secretome are scored based on the HTPS Motif Score. The distribution of HTPS Motif Score is filtered using a cut-off of 0.1 FPR using as true positive the cleavages deposited in MEROPS. In the second filtering step, the prediction of amino acid accessibility is used to identify proteases-accessible substrates. In the third step a protein level filter is applied to exclude proteins, for which the concentration in the secretome was not determined (ProteinAtlas database) and/or are not co-cited with the investigated protease. In the final matrix, all proteins are ranked based on the number of identified substrates, protein co-citation and protein abundance. In the right part of the panel, the search space reduction across the three filtering steps (expressed as number of potential substrate sequences for α-Thrombin) is shown. **b** α-Thrombin HTPS Motif Score distribution (light red) and true positive distribution (light blue) calculated from the positional enrichment of each amino acid of all secretome proteins against the HTPS Motif Score. **c** Distribution of the HTPS Motif Score of α-Thrombin generated for all proteins of the secretome. Annotated α-Thrombin physiological substrates are depicted in blue. The HTPS Motif Score cut-off is highlighted by a dashed red line. **d** A Receiver-Operator Curve (ROC) of α-Thrombin to evaluate the performance of the filtering step. **e** Matching of predicted and detected cleavage events in case of α-Thrombin acting on C3 (Complement component 3). Identification of α-Thrombin cleavage sites in C3 was performed with a simplified di-methylation reaction from TAILS protocol and the detected cleavages are shown on the upper part of the picture. The lower part shows the positional HTPS Motif Score distribution along the C3 protein sequence. Only the positive HTPS Motif Scores are shown in the plot. **f** The distribution of HTPS Motif Score for validated C3 cleavages (red, median = 2.86, $n = 17$) and for all positions (gray, median = −1.95, $n = 1,663$). The boundaries of the box plot correspond to the quantiles Q1 (25%) and Q3 (75%). Lower and upper whiskers are defined by Q1 −1.5IQR and Q3 +1.5IQR. The calculated $p$-value (paired Wilcoxon test) corroborates the validity of HTPS Motif Score as a proxy to select potential protease substrates for further biochemical validation. **g** The distribution of the number of proteins identified by the filtering strategy generated by unique proteases or shared by multiple proteases. **h** The distribution of the number of proteases predicted to cleave the corresponding protein substrates demonstrates that coagulation cascade proteases are generally highly promiscuous proteases. **i** Sankey diagram showing the distribution of proteases and protease inhibitor classes across the identified candidate substrates. **j** Bar plot of the number of candidate substrate sequences identified for the 41 serine proteases included in the final filtered target list. Proteases associated with complement and coagulation are shown in blue and red, respectively, and main represented families (based on Panther database) are reported as colored squares for Tryptase-like (yellow), Kallikrein-like (green) and Trypsin-like (red) proteases. **k** The network of coagulation proteases and their substrates generated from protein-protein interaction database (simple connection) (BioGRID v.3.6.1.8.2) and MEROPS substrates database (arrow). Protein substrate information generated from the HTPS data by the previously described filtering steps was superimposed onto the network (black edges) demonstrating that HTPS data can comprehensively recapitulate protease-substrate relationships of the coagulation cascade. Source data are provided as a Source Data file.

and important for crosstalk between the coagulation and complement systems[76], was identified in the top 5 rank as potential physiological substrate of α-Thrombin. To validate the predicted cleavage sites, we performed a cleavage assay of C3 with α-Thrombin and identified the cleavage sites using a simplified reductive di-methylation TAILS workflow[16]. Subsequently, the thus identified cleavage sites were matched with the predicted sites (Fig. 6e, Supplementary Data 9). Remarkably, we could confirm 5 out of 11 predicted cleavage sites and show how HTPS filtering approach based on residual exposition and protein features (abundance, co-citation, number of cleavages detected) can be applied to the protease specificity data obtained from a native cell lysate to predict cleavage events in an extra-cellular environment. Moreover, all identified cleavage sites for C3 protein (17 in total) were predicted with an HTPS Motif Score > 2 (the median value for the protein was negative), indicating that it can successfully map cleavages product in the top 5% hits (Fig. 6f).

Of note, we also found that using this filtering strategy, most target sequences were unique to specific proteases and only a few were shared among all six (Fig. 6g). Interestingly, the protein substrates displayed an opposite trend: only a few proteins were targeted by a single protease, and the large majority was potentially a target of several or even all of them. The high number of target sequences carried on average by each target protein seems to explain to a significant extent this observation (Fig. 6h).

Next, we asked which processes and functions were enriched among the proteins targeted by the blood cascade proteases. We used DAVID[77] to calculate the enrichment against the secretome background and found that, in line with our expectations, proteins involved in complement activation and fibrinolysis were enriched among the potential targets, with the class of serine-type proteases being most significantly represented in this subset (Supplementary Fig. 12l). A Sankey diagram (Fig. 6i) shows indeed that serine proteases were the main substrates of the investigated blood cascade proteases, which displayed generally similar connectivity also with other proteases and protease

inhibitor classes. By plotting the number of target sequences for all 41 serine proteases found in blood, a number of additional trends emerged. All proteases that were part of the complement and coagulation pathways were among potential candidates, while about a third of the proteases, especially kallikrein-like and tryptase-like, had no target sequence identified after filtering (Fig. 6j). Collectively, the observations about the correctness of scoring and recall of known substrates as well as enrichment of relevant biological processes and functions, indicate that this strategy is able to recover potentially relevant physiological substrates of investigated proteases.

Finally, we combined the knowledge about protease substrate relationships deposited in MEROPS and the information about protein-protein interactions deposited in BioGRID[78] (v.3.6.1.8.2) in a single network and overlaid it with the data retrieved with our method. Remarkably, as shown in Fig. 6k, our analysis was able to capture the large majority of previously known protease-substrate relationships and protein-protein interactions in an entirely data-driven way. We thus defined a strategy to generate context-relevant substrate predictions from HTPS-experimental results obtained in generic systems. This strategy allowed the reduction of the sequence search space by more than 3 orders of magnitude and was able to isolate biochemically, structurally and biologically plausible and thus likely relevant protease-substrates relationships.

## Discussion

Here we describe and benchmark a high-throughput protease screen (HTPS) and demonstrate its performance with selected applications for protease research. We characterized 15 proteases under physiologically relevant conditions and, excluding results from proteases used for standard proteomic workflows such as Trypsin and Lys-C, identified more than 160,000 unique substrate cleavages, thus substantially expanding the currently available protease knowledge base. The protocol is simple, scalable, robust, easy to parallelize for multiple conditions (reduces batch effects), avoids any chemical modifications or labeling and, as few

biochemical steps are required (no enrichment or depletion), it reduces sample loss. Importantly, the simplicity of the FASP-based HTPS protocol is also suited to incorporate parallel or sequential digestion steps, which might be beneficial for studying proteases that generate lower cleavage numbers. A suite of publicly accessible scripts that support the analysis of the generated data complement the wet lab protocol. As an example, the screening of nine coagulation proteases in triplicates under three different conditions (with NaCl, ChCl and LiCl) required typically 2–5 μg of individual tested protease, the native cell lysate of a single 15 cm dish (5 mg of total proteins) as substrate sample, and could be carried out in only half day of bench work and 2 h of MS acquisition time per sample. The benchmarking of the method with standard proteomic proteases, WN NS3 protease and metalloproteases has produced two main conclusions. First, HTPS is able to recapitulate accurately protease specificity with a performance comparable to other methods. Second, HTPS does generally lead to the identification of vastly larger numbers of substrate peptides identified per protease, in comparison with most of the other methods so far used for protease characterization. Furthermore, the highly parallel setup reduces batch effects and increases the method throughput. It can also simultaneously recover prime and non-prime substrate specificity (besides of DIPPS[21], PICS[79] and ChaFraTip[22]), but does so in native conditions.

We demonstrated the microscale and high-throughput capabilities of HTPS by applying the workflow to a set of coagulation cascade proteases and detect specificity features for activated α-, β-, γ-Thrombin, aFVII, aFIX, aFX, aFXI, aPC and PLG. Here, the high numbers of detected cleavages allowed us to characterize the minor distinguishing features between these closely related proteases and group them according to their cleavage specificity and cleavage entropy. Furthermore, we were able to recapitulate from our proteomic data the known specificity differences between two isoforms of Thrombin (α- and γ-), which further demonstrated the sensitivity of the screen. The large number of cleavage events identified per measurement allowed us to investigate the effect of cofactors on protease activity and the allosteric effect of Na$^+$ on their activity and specificity with great sensitivity. We obtained results that confirm the mechanisms of allosteric regulation for α-Thrombin[57,64] and aFX[62] and expand our knowledge to other blood proteases for which so far mechanisms of allosteric regulation with Na$^+$ were not extensively described. This demonstrated that differential specificity and entropy profiling can be used to identify restraints to model conformational changes. It is also important to note that allosteric effects are typically investigated with fluorescence anisotropy, biochemical or structural studies, which often require high amounts of proteases (e.g. in mg range for protein crystallography). In contrast, in its current implementation HTPS analyses are performed with proteins in their native fold, require less than 1 μg of protease per assay, and further downscaling can be envisioned.

The translational value of HTPS is perhaps best illustrated in the context of designing sensitive tools for detection of protease activity. We used HTPS data to design synthetic peptides and show that they were cleaved by their respective proteases, demonstrating that positional substrate preferences detected with the protocol can translate into tools for detecting protease activity. This is useful, especially in case of poorly characterized proteases where a fast and simple design of a substrate can assist further protease characterization steps. An exemplary application of this concept could be the design of test substrates to characterize proteases of a newly emerging virus as exemplified by the profiling of WN NS3 viral protease. Moreover, large protease datasets could possibly serve as a hypothesis-generator for targeted assays[80] and for spike-in assays used for detection of

protease activity[81] as recently demonstrated for asparaginyl endopepdidase[82]. Furthermore, protease datasets could support the development of assays that serve as sentinels to follow biological processes in a high-throughput fashion[83]. It must be borne in mind, however, that HTPS is limited to amino acids that naturally occur in proteins in comparison to synthetic peptide libraries. When designing specific substrates for proteases, especially if the target group are closely related proteases, including non-natural amino acids to protease screens is beneficial and can provide another level of selectivity[84].

As a final, highly relevant application, we show that the large number of identified protease cleavages in near-native conditions can be exploited to predict relevant substrates in systems orthogonal to those experimentally used. Here, a simple computational filtering framework, largely based on HTPS-results, combined with readily available orthogonal information, was capable to retrieve a large number of physiologically relevant relationships. Among these, we validated with an orthogonal technique the predicted α-Thrombin cleavages on the C3 complement protein demonstrating that HTPS cleavage motifs obtained from a native cell lysate can be used to generate hypotheses on physiological substrates. While HTPS is not intended to directly study in vivo proteolysis, the method can be employed to generate hypotheses on as yet unexplored connections. The substantial pool of substrates/cleavage events identified with HTPS may play in the mid-term also an important role in bringing machine learning approaches to protease research and improve the performance of tools readily used for prediction of protease substrates. Recent developments of tools like iProt-Sub, that can predict cleavages in protein substrates, demonstrated the importance of having a detailed and representative cleavage dataset for the investigated proteases to retrieve their specificity features and thus construct better models that could enable proteome-wide prediction of protease substrates[85].

To conclude, we introduce a proteomic tool for protease research, which we dub HTPS. We believe it could be readily applied for large-scale de-orphaning of proteases, systematic comparison of their specificity and cleavage entropy, identification of potential physiological substrate candidates for validation in biochemical assays, as well as generation of substrate reporters to investigate protease activity and structural rearrangements. Further improvements, including adoption of more sensitive MS, shorter LC gradients and scaling down of the starting material, will make profiling of the entire human protease repertoire across different conditions a goal within reach, as only ~9 sets of experiments would, in principle, be sufficient to profile it in triplicates on a 384-well format.

## Methods

**Proteases used in the study.** All proteases used in this study were purchased from commercial vendors, Trypsin (V5111), Glu-C (V1651), Asp-N (V1621) and Chymotrypsin (V1091) from Promega (USA) and Lys-C (125-05061) from Wako (Japan). The blood cascade proteases α-Thrombin (HCT-0020), β-Thrombin (HCBT-0022), γ-Thrombin (HCGT-0021), Factor VIIa (HCVIIA-0031), IXa (HCIXA-0050), Xa (HCXA-0060), XIa (HCXIA-0160), Plasmin (HCPM-0140) and activated Protein C (HCAPC-0080) were purchased from Hematologic Technologies, Inc., (USA). Recombinant human MMP2 (902-MP-010) and recombinant West Nile Virus NS3 Protease Protein (2907-SE) were purchased from R&D systems (USA). Recombinant human MMP3 (SRP7783) was purchased from Sigma Aldrich (Germany). The active concentration of coagulation proteases used in this study was determined by active site titration using the irreversible stoichiometric inhibitors TPCK (Sigma Aldrich), PPACK (Hematologic Technologies) and GGACK (Hematologic Technologies), according to standard active site titration protocols[86].

**Cell culture and preparation of native cell lysates.** Human Embryonic Kidney 293 cells (HEK293, ATCC CRL-1573) were grown under standard conditions in DMEM (Gibco) supplemented with 10% FBS (BioConcept), 1% glutamine (Gibco), and 1% penicillin/streptomycin (Gibco) at 37 °C in a humid incubator at 5% $CO_2$.

When the cells reached 90% confluence, they were detached from the plate with a jet of PBS (Gibco) and washed twice with PBS. For lysis, we used mild lysis conditions with HNN buffer (50 mM HEPES, 150 mM NaCl, 50 mM NaF, pH 7.8) supplemented with 0.5% NP-40 and protease inhibitor cocktail according to manufacturer's recommendations (Sigma Aldrich) as described elsewhere[87]. Afterwards, the lysate was centrifuged at 14,000 $g$ for 15 min to remove any non-soluble material and the buffer was exchanged for 20 mM Ammonium Bicarbonate pH 7.8 using a filter device with molecular weight cutoff of 10 kDa. Standard BCA protein assay was used to determine the total protein concentration (Thermo Fischer Scientific), the concentration of the standardized lysate was set to 1 mg/ml and stored at −80 °C until used for the digestion assays.

**Protease digestions and sample preparation**. All protease digestions were performed in 96FASP plates with MWCO 10 kDa (Acroprep Advance™) by adapting a 96FASP sample preparation protocol for protease digestion under native conditions[28,29]. The first step was to wash the filter units to remove any residuals. For this 100 μl of 20 mM Ammonium bicarbonate pH 7.8 were added to the wells and the plate was centrifuged at 1,300 $g$ for 10 min before repeating this step once more. Afterwards, native cell lysate standardized in 20 mM Ammonium bicarbonate pH 7.8 was added at a final 50 μg of total protein per well and mixed with the investigated proteases at 1/50 [E]/[S] ratio. The samples were incubated at 37 °C for 4 h and collected by a 15 min centrifugation at 1,300 $g$ in a low binding 96-well conical plate. The collection step was repeated by adding 100 μl of MS-grade water. The fractions were transferred to low-binding tubes (Eppendorf) and concentrated on the SpeedVac to complete dryness. The samples were stored at −80 °C until analysis. Before analysis, the samples were re-suspended in 20 μl of MS-grade water with 0.1% formic acid and the peptide concentration was determined with Nanodrop UV spectrometer. The sample concentration was adjusted to 1 μg/μl with water containing 0.1% formic acid.

**Allosteric effects of Na⁺ on blood cascade proteases**. To investigate the effect of Na⁺ on the proteases of the blood cascade we performed the assay in presence of 0.2 M of NaCl or choline chloride (ChCl) as previously reported[60]. In some of the assays, LiCl was used as a control at 0.2 M. In selected assays we also included Tissue Factor (Recombinant Tissue Factor, RTF-0300) or Thrombomodulin (Rabbit Thrombomodulin, RABTM-4202), both purchased from Hematologic Technologies (USA). In case of cofactors, we incubated the cofactor and the protease for 30 min at 10 °C using a 10-fold excess of the respective cofactors, before using the protease for the HTPS screen. We performed the digestion experiments with blood cascade proteases under both conditions for 2 h at 20 °C in 96-well plates and collected the peptides as previously described. Importantly, all allostery assays were performed at pH 7.4. Additionally, before the LC-MS/MS analysis we performed a desalting step of the samples with C18 UltraMicroSpin columns according to the manufacturer's protocol (The Nest group, USA). The dried peptide samples were re-suspended in 0.1% FA water at a concentration of approximately 1 μg/μl.

**LC–MS/MS analysis**. The LC–MS/MS analysis of the protease-digested samples was performed on an Orbitrap Elite (Thermo Fischer Scientific) interfaced with an Easy 1000 nano-LC unit (Thermo Fischer Scientific), coupled online with the nano-electrospray. The LC–MS/MS was operated with the Xcalibur software package (Thermo Fischer Scientific). For the analysis, 1 μg of sample was loaded directly on the analytical column (Acclaim PepMap™ RSLC, 75 μm × 15 cm, nanoViper C18, 2 μm, 100 A, Thermo Fischer Scientific). The flow rate on the nano-LC was set to 300 nl/min and the peptides were chromatographically separated with a 5–35% 120 min linear acetonitrile/water gradient in 0.1% formic acid. During the entire run, the MS spectra were acquired in the Orbitrap in positive ion mode with 2.0 kV voltage in the mass range of 350 to 1,600 m/z, set to the profile mode and a resolution of 120,000 at 400 m/z. For peptide fragmentation, a CID fragmentation method with normalized collision energy 35 was used and the MS/MS spectra were obtained from the 15 most intense precursor ions from the full MS spectra. During the entire run, precursors with repeat count of 1 were dynamically excluded for 30 s. Precursors with charges +2, +3 and +4 were considered and the MS/MS spectra were recorded in the ion trap analyzer in the centroid mode with normal scan rate and standard settings.

The analysis of the allostery samples was performed on an Orbitrap Fusion (Thermo Fischer Scientific) interfaced with an Easy 1000 nano-LC unit (Thermo Fischer Scientific) and operated as described previously. 1 μg of sample was loaded directly on the analytical column made in house (75-μm inner diameter; New Objective) with ReproSil-Pur 120 A C18 1.9 μm (Dr. Maisch GmbH) as stationary phase. The flow rate was set to 200 nl/min and the peptides were chromatographically separated with a 5–25% 90 min acetonitrile/water gradient in 0.1% formic acid. The data acquisition mode (data-dependent acquisition) was set to perform a cycle of 3 s with high resolution MS ($R = 30,000$, AGC = 50 ms) and MS/MS ($R = 60,000$, AGC = 54 ms) in the Orbitrap analyzer. During the entire run, the MS/MS spectra were acquired in the Orbitrap analyzer in the mass range of 350 to 1,650 m/z; precursors with charges 2–7 and intensity higher than $2*10^4$ were selected for fragmentation (HCD, NCE = 28). The dynamic exclusion window was set to 30 s. For quality control a standard sample of iRT peptides

(Biognosys AG, Switzerland) was injected after each analyzed HTPS sample triplicate. The retention times of iRT peptides and the corresponding MS2 intensities were compared with Skyline[88].

**Database searches and abundance-focused library generation**. The raw data was searched with MaxQuant[31] (version 1.5.2.8) using the human UniProt database (Homo sapiens, UniProt release October 2018, 20,382 entries) and the in-house generated abundance-focused HTPS_DB.fasta database (2,557 entries). For generation of the HTPS_DB.fasta abundance-focused database, we combined the lists of proteins that were identified in the samples after treatment with Trypsin, Lys-C, Asp-N, Glu-C and Chymotrypsin. For specific database searches we used standard MaxQuant settings[89], for searches without a defined enzyme specificity we set the digestion mode to unspecific and the maximal peptide length to 40 AA as described elsewhere[21]. Our searches considered only two natural PTMs, acetylation of N-termini (+42.0106 Da) and the oxidation of methionine (+15.9949 Da) as variable modifications. First search peptide mass tolerance was 20 p.p.m. and main search peptide mass tolerance was 4.5 p.p.m., as set by default. MS/MS match tolerance was set to 0.5 Da. For the peptide identification via peptide-spectrum matching the FDR was controlled with a standard target-decoy approach[89]. A 1% peptide FDR was applied at PSM level and only peptide hits with a PEP score ≤0.05 and a score >40 were retained for further analysis.

The final list of proteins was the union of proteins identified in the respective samples and we included only proteins with a global protein PEP ≤ 0.01 into the final database. Potential contaminants were excluded from the subsequent data analysis.

**Data analysis and visualization**. Data analysis was performed in R (version 3.4.3) using the workflow deposited on Github (https://github.com/anfoss/HTPS_workflow, https://doi.org/10.5281/zenodo.4484341) under MIT license. Briefly, the script recovers the cleavage sequences from the identified peptides and transfers them into a positional matrix (amino acids upstream the cleavage site occupy P8-P1 position and amino acids downstream P1'-P8' position). A frequency matrix is generated counting the abundance of amino acids per position and normalized for all identified peptides. To harmonize the multivariate protease specificity data, the positional occurrences of amino acids are converted into protease frequency matrices. In parallel, a random frequency matrix of the same size is generated by sampling the same number of amino acids as contained in the frequency matrix from the natural distribution of amino acids in HTPS_DB.fasta. The proteases were first compared in terms of numbers of generated cleavages under different tested conditions. For visualization of the specificity, we used the iceLogo program[38] with the threshold of significance $p$-value set to 0.01, respectively. To compare proteases in terms of significantly different positional features, a two-side paired t-test was employed to evaluate similarity between frequency matrices of different proteases or differential frequency matrices of the same protease under different conditions and thus to evaluate the similarities/differences between the tested proteases/conditions. The evaluation and comparison of substrate specificity for MMP2 and MMP3 with PICS, TAILS and DIPPS data was performed by adapting the workflow used for HTPS. Identified cleavages or peptides reported in the studies[21,43–45] were used to generate the frequency matrix and the specificity enrichment using as a control the random distribution of amino acids from HTPS_DB.fasta database. For conditional protease comparison, we took the significant ($p$-value < 0.01) enrichment of amino acid per position compared to the random distribution in presence of NaCl and ChCl, compared the folds of change and report the significant changes according to $p$-value. The calculation of cleavage entropy was performed as a Shannon entropy calculation[34]. The block entropy calculation was performed as described elsewhere[35].

**Spike-in octapeptides and fluorescent substrates for α-thrombin and factor X**. The octapeptides GIPRAAGD (α-Thrombin) and GIGRRIAE (aFX) were synthesized by the solid-phase method using the 9-fluorenylmethyloxycarbonyl (Fmoc) strategy on a model PS3 automated synthesizer from Protein Technologies International (Tucson, AZ), according to a standard protocol described elsewhere[90]. The crude peptides were subsequently purified by RP-HPLC on a C18 analytical column (Grace-Vydac, Hesperia, CA) and analyzed by MS with a data-dependent acquisition (DDA) approach. In order to determine the linear response range for the proteases, the two peptides were tested from 100 μM to 10 pM. To confirm the octapeptide cleavage we incubated 10 μM of the peptide with a 10 nM final concentration of proteases from 0–240 min. For the analysis, 1 μg of sample was loaded directly on reverse phase column (75 μm × 15 cm, packed with Magic C18 3 μm resin) and the peptides were separated with a 5–35% 20 min linear acetonitrile/water gradient in 0.1% formic acid with a flow rate set to 300 nl/ml, using a Proxeon EASY-nLC II chromatography system (Thermo Fischer Scientific). The acquisition started with sample injection. The MS1 quantification of selected reporters was performed on an Orbitrap XL (Thermo Fischer Scientific) in positive mode with 2.0 kV voltage in the mass range of 150 to 1,200 m/z in the profile mode at a resolution of 60,000 at 400 m/z. The measurement was performed using 1 μl of the standardized sample spiked with iRT peptides (Biognosys AG) at 1:20 and proteolyzed BSA at 0.1 mg/ml as carrier. We manually integrated the precursor isotope peaks (M, M + 1, M + 2) using Skyline software[88] of GIPRAAGD (378.70

m/z), GIPR (221.64 m/z), AAGD (333.14 m/z) for α Thrombin and GIGRRIAE (436.26 m/z), GIGR (201.63 m/z), RIAE (488.28 m/z) for aFX.

The fluorescent substrates for α-Thrombin (zGIPR-AMC) and for aFX (zGIGR-AMC) were purchased from Biomatik (USA) and selectivity was tested in a standard protease screen as described elsewhere[69]. All measurements were performed in 20 mM Ammonium bicarbonate pH 7.8 supplemented with 200 mM NaCl. Where applicable, we also determined the corresponding $k_{cat}/K_M$. The substrate concentration range in the assays was 1 μM–200 μM, the protease concentration range was 1–5 nM. We monitored the increase of fluorescence intensity with a Tecan infinite 2000 Pro plate reader (Tecan, Switzerland) in continuous mode (excitation at 370 nm, emission at 460 nm) and calculated the corresponding $k_{cat}/K_M$ values as earlier described[69] using GraphPad (Prism).

**In silico data analysis**. Molecular docking was performed with HPEPDOCK web server[91], starting from the structures with the water molecules removed and inhibitor-free for α-Thrombin (1ppb [https://doi.org/10.2210/pdb1ppb/pdb])[92] and aFX (1g2l [https://doi.org/10.2210/pdb1g2l/pdb])[93] and the two octapeptides. The software generated 3D structure models for the given sequences of peptides using the implemented MOPEP program, which considers peptide flexibility. Simulations were run with default parameters, without introducing any geometric or energetic constraints. One hundred poses were generated and ranked according to the CAPRI criteria[94]. The most acceptable prediction was selected for the data analysis. PyMOL software (v. 0.99rc6) was used for visualization of the docking results.

**Identification of candidate substrates**. To identify physiologically relevant protease substrates for validation in biochemical assays we applied three filtering steps. In the first filtering step, we calculated a motif score for all the combination of amino acids (280,771) for all secretome proteins (secretome database from Protein Atlas[73,74]). The motif score for each protease analyzed (Trypsin, α-Thrombin, aFVII, aFIX, aFX, aFXI, PLG and aPC) was calculated from the sum of significant fold changes associated with the respective residues compared to a random distribution generated from HTPS database. To evaluate the performance and to identify a cut-off at 1% FPR we generated a receiver operator curve (ROC) using as true positive the annotated MEROPS substrates (release 12.1)[23]. The identified substrates were further filtered based on the prediction of amino acids exposition using JPred4 tool[75] (http://www.compbio.dundee.ac.uk/jpred/). For this filtering step, we split the set of proteins in pieces of 750 amino acids with overlap fragment of 20 amino acids, we calculated the accessibility using the intermediate score "JNETSOL_5" and we filtered all substrates that were not buried ($n = 8$). In the last step, we applied a protein-based filtering step. In this step we removed proteins for which the concentration in blood was not reported (ProteinAtlas, Secretome[73,74], https://www.proteinatlas.org/) and/or were not co-cited with the studied individual coagulation protease in PubMed (https://www.ncbi.nlm.nih.gov, ftp://ftp.ncbi.nlm.nih.gov/gene/DATA/gene2pubmed.gz). Furthermore, proteins were scored by multiplying the inverse of ranking position for (i) co-citation frequency, (ii) number of identified protease substrates, (iii) concentration in the blood. GO enrichment for Biological Process and for Molecular Function was performed using DAVID tool[77] (v6.8, https://david.ncifcrf.gov/) using the human secretome (ProteinAtlas) as background. Protease substrate network was generated using Cytoscape (v.3.8.0)[95,96], combining data of reported protein interaction in BioGRID (v.3.6.1.8.2)[78] and substrates from MEROPS (v.12.1)[23] for all coagulation protease.

**Identification of thrombin cleavages on complement factor C3**. Purified human complement protein C3 (A113, Complement Technology, Inc., USA) was used as a substrate in a cleavage assay with α-Thrombin. 10 μg of C3 protein was exposed to the protease at 1/100 [E]/[S] ratio and the reaction was incubated at 37 °C for 2 h in 50 mM HEPES buffer pH 7.4 supplemented with 200 mM NaCl. As a control, the protein sample was incubated under the same conditions without the protease. After the proteolysis, the reaction was terminated by heat-inactivation of the protease. The sample was mixed at 1:1 ratio with 100 mM HEPES pH 6.5 and reductive di-methylation of free N-termini was performed by adapting the steps from the di-methylation reaction from TAILS protocol[16]. Briefly, formaldehyde (Sigma–Aldrich) and NaCNBH₃ (Sigma–Aldrich) were added to the post-proteolysis/incubation samples in 2:1 ratio to reach a 20 mM and 10 mM final concentration, respectively. The reaction mixture was incubated for 16 h at 37 °C before the reaction was stopped by diluting the reaction mix 1:5 in 50 mM ammonium bicarbonate pH 7.8. To digest the labeled protein fragments, Trypsin was added to the reaction at 1:50 ratio and the reaction was allowed to proceed for 16 h at 37 °C. The reaction was terminated by acidification with formic acid to a final 0.5% and the peptides were recovered with the use of a standard C18 cleanup procedure (The Nest group, USA). The identification of α-Thrombin cleavages on C3 protein was performed by specific database searches with MaxQuant[89] as earlier described, where di-methylation of N-termini and Lysine side chains (+28.0313 Da) were considered as variable modifications. The labeled N-termini present in the α-Thrombin-treated samples (PEP < 0.01) but absent from the negative controls were considered as α-Thrombin cleavage events and compared to the cleavage sites predicted with HTPS Motif Score.

**Reporting summary**. Further information on research design is available in the Nature Research Reporting Summary linked to this article.

## Data availability
The data is deposited to ProteomeXchange Consortium via the PRIDE partner repository[97] with identifiers PXD018976, PXD020320, PXD022959, PXD022971, PXD022972 and PXD022973. Source data are provided with this paper. All protein structures referred to in this study were obtained from PDB (https://www.rcsb.org/), with the accession codes 1ppb [https://doi.org/10.2210/pdb1ppb/pdb] and aFX (1g2l [https://doi.org/10.2210/pdb1g2l/pdb]). The data of this study is available within the paper and the corresponding supplementary information. Additional information or other potentially relevant data are available upon request from the corresponding author. Source data are provided with this paper.

## Code availability
R scripts to analyze the data and reproduce the reported data analysis are available at https://github.com/anfoss/HTPS_workflow and https://doi.org/10.5281/zenodo.4484341 under MIT license[98].

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

## Acknowledgements
R.A. and M.V. are supported by the Swiss National Science Foundation through grant # SNSF 31003A_166435 to R.A. and by the European Research Council (ERC) through grant 20140AdG 670821. M.G., F.F. and F.U. acknowledge support by the IMI project ULTRA-DD (FP07/2007-2013, grant no. 115766). U. adK. acknowledges support by a Novo Nordisk Foundation Young Investigator Award (NNF16OC0020670). S.G. and B.W. acknowledge support by the Personalized Health and Related Technologies (PHRT) strategic focus area of ETH. This work was also supported by a Grant from the CaRiPaRo Foundation Excellence Research Project 2018 BPiTA n. 52012 to V.D.F.

## Author contributions
M.V., F.U. and R.A. conceived the study. M.V. and F.U. performed the experiments, F.U., M.V., R.C. and A.F. analyzed the data. A.F. and F.F. contributed to the experimental design. L.A. and V.D.F. synthesized the substrate peptides and provided valuable input for experiments. U. adK. provided information for MMPs and valuable discussions. S.G. and B.W. provided access to the PHRT platform for the allostery experiments. M.V. and F.U. wrote and R.A. revised the manuscript. M.G. provided funding. All authors edited the manuscript. R.A. supervised the project and provided funding.

## Competing interests
The authors declare no competing interests.
