## [Peer Review File · Nature Communications]

REVIEWER COMMENTS

Reviewer #1 (Remarks to the Author):

This study reports a simple proteomic method to characterize protease specificity using products from native lysates obtained under physiological conditions and analyzed by mass spectrometry. The Authors claim that "the method is significantly faster, cheaper, technically less demanding, easily multiplexed and produces accurate protease fingerprints". Particular attention is devoted to proteases of the blood coagulation cascade for which the Authors obtain "substrate profiles of unprecedented depth". Attention is also paid to characterize "entropy" and "allosteric changes" of the protease and the prediction of physiological substrates.

Overall, I find the study interesting as a methodological tool, but less so as a conceptual advance to the field. My suggestions for revision are largely limited to the coagulation proteases and some interpretations.

1. It is unclear what the Authors mean by "entropy". For sure they do not refer to a thermodynamic quantity but they also provide no measure of this quantity, context and practical meaning.

2. Blood coagulation proteases, especially the vitamin K-dependent ones, are lousy enzymes without their physiological cofactors. Were cofactors used in the assays? For example, the activity of thrombin toward protein C without thrombomodulin is nearly unmeasurable. Ditto for the intrinsic activity of FVIIa without tissue factor. It is highly problematic to establish entropy and allosteric effects when working with poorly active enzymes. To that end, were enzymes titrated for activity?

3. The "allosteric" effect of Na⁺ needs attention. In the case of thrombin, Na⁺ does not change specificity from fibrinogen to protein C (Figure 4A is decades old and potentially misleading), but simply increases the activity toward most substrates. The change in "specificity" from fibrinogen to protein C is produced by thrombomodulin binding and is entirely a steric effect that excludes one substrate and let the other one in. Whether Na⁺ binding changes the specificity profile of thrombin remains to be seen, especially in the presence of thrombomodulin. Similar arguments apply to FVIIa, FIXa, FXa and activated protein C in the presence of tissue factor, FVIIIa, FVa and protein S.

4. The strategy of characterizing the Na⁺ effect from studies that compare NaCl vs ChCl definitely works for thrombin but not in general. It has been known for two decades that FXa is specifically inhibited by ChCl and quaternary amines. Other coagulation proteases may be inhibited by ChCl as

well. The "no Na+" conditions should be produced by either eliminating Na+ (no constant ionic strength) or adding LiCl as "inert" salt.

5. Assays were carried out under physiological conditions, but the pH used was 7.8. Why not 7.4? Also, Na+ binding to coagulation proteases is weak and very temperature dependent. For some proteases, 200 mM NaCl at 37 C means very little Na+ bound. If the goal is to establish the effect of Na+ on specificity, conditions should be used where binding of Na+ is maximized, increasing NaCl and working at 20-25 C.

6. It should come as no surprise that a protease may have a specificity profile much wider than suspected from its known panoply of physiological targets. Does that mean that other unknown targets of physiological interest may be discovered? And would that be facilitated by known cofactors? The Authors may want to comment on that.

Enrico Di Cera

Reviewer #2 (Remarks to the Author):

The manuscript by Uliana and Vizovisek describes an easy-to-execute, high-throughput screening method to profile protease specificity. Overall, I like this work because of its simplicity and some smart tricks, such as using a search space that is tailored to the used sample. In short, the method is based on treating a cell lysate with proteins in their native conformation with a protease of interest, followed by filtering over a 10 MWCO filter, which isolates the digested peptides, which are then directly analyzed by LC-MS/MS.

At first, the authors benchmark their method against several proteases commonly used for digestion in proteomics, as well as two MMPs. Then, they use different proteases from the blood coagulation cascade to profile specificity and investigate the role of sodium binding as allosteric regulator. Furthermore, they use the found protease specificity information to design selective protease substrates and search for native substrates.

I think that the experimental design is done well, the data analysis is also performed well and conclusions are based on the data. The authors provide scripts online that will help other laboratories to implement this workflow.

Because of the simplicity of the sample preparation, I think the method will be easily transferable to other laboratories and therefore attract broad interest of researchers in the field of proteolysis. I

therefore think that Nature Communications is an appropriate journal, and I recommend publication of this manuscript after a minor revision as detailed below:

Comments:

1. P3 line 57, and also P20 line 489: the authors state that HTPS is used in near-native conditions. I don't like this term. What is a near-native condition? In section 2.3 blood coagulation cascade proteases are used on HEK cell lysates. As blood coagulation proteases usually only encounter blood serum, this really cannot be called near-native conditions (the authors state this also in their results section). Furthermore (although I don't want to be too nitpicking), the protein concentration is 1 mg/mL, which is approximately a factor 200 lower than native conditions in HEK cells. HEPES? Not native etc. I think that a better phrasing would be: HTPS is used on lysates with proteins in their native fold (and I guess that is what the authors mean). But I anyway think that this term should not be emphasized too much, because I doubt if it really matters for the results. The strength of the study is: an amazingly simple method for protease substrate specificity profiling. Whether the substrate proteins are present in their native fold or in a different conformation, I don't expect that this would have a big influence on the data.

2. P5, line 89: note that ChaFraTip can also be used on non-digested proteins/cellular samples (see: Shema, Zahedi et al, Mol Cell Proteomics 2018)

3. P7, line 145: "which retains undigested proteins and the added protease and supports recovery of the cleavage products in the flow-through." This assumes that all proteolytic cleavages will lead to cleavage products that are lower than 10 kDa. I agree that this will indeed be the case with the test proteases trypsin, GluC, etc. But I am convinced that this is not the case for all proteases and all substrates: Sometimes a very small part (propeptide) is cut from a protein and it virtually retains its original MW. Or in case of very limited proteolysis, fragments would be larger than 10 kDa. In the discussion (line 512) the authors speculate about viral proteases, but these usually cut one large viral protein into large fragments (> 10 kDa) that form functional proteins. I think the authors should briefly mention/discuss this in their introduction or discussion, and also remove their statement about viral proteases in the discussion, because I don't think this would be an appropriate application (especially if you would work on with the native viral protein). On a human cell lysate (folded or denatured), it may lead to sufficient cut sites to generate small enough peptides to be isolated in the flow through.

4. Block cleavage entropies -> I have no idea what this really means, and I think that it will have no meaning to a reader, if it is not explained in the figure caption of Figure S6. I briefly took a look at Reference 35, and as far as I understand, these numbers reflect if the protease relies on the occurrence of specific successive amino acids in small blocks along the cleavage site. I think the authors should mention this and also indicate what the numbers on the y-axis mean (i.e. what is

their implication if they are higher or lower). There also seems to be a mistake in the figure caption of S6, because the block entropy values are not given per position (as written in the caption), but per block (as indicated in the graphs). Please indicate what B4, B3, B2 etc stands for.

5. P16, line 396: S2238 does not only have a non-natural P3 D-Phe, but also a P2 Pip (“homoproline”) as non-natural amino acid, likely affecting the selectivity. Please mention.

Reviewer #3 (Remarks to the Author):

The authors present a method for high-throughput profiling of protease cleavage site specificity based on a standard in solution digest of isolated, non-denatured proteomes under semi-native conditions, essentially substituting trypsin by the test protease of interest in filtered-aided sample preparation. In “High-Throughput Protease Screen” (HTPS), cleavage products are isolated by spinning through a 10 kDa filter, followed by mass spectrometry-based identification. Unspecific peptide sequence matching was constrained using a database limited to proteins identified by the same method using standard digestion enzymes. This resulted in the identification of several hundred to thousands of cleavage sites for each tested protease. These rich datasets were used for an in-depth characterization of cleavage site specificity using a combination of previously established data analysis routines, including analysis of preferred substrates using iceLogos, calculation of cleavage entropy and block entropy as comparative measures for specificity and cooperativity of neighboring sites.

The serine proteases of the blood coagulation system were chosen for proof of concept analysis, and the results obtained using HTPS recapitulated prominent features documented in the literature. Based on the HTPS results, two fluorescently labeled tetrapeptide substrates were designed to successfully exploit a differences in P2 subsite selectivity between aFX and thrombin, respectively. Finally, candidate physiological substrates were predicted based on the match in specificity, followed by a scoring scheme that utilizes literature knowledge such as predicted co-localization, co-expression, and co-citations.

HTPS appears technically sound and will be a useful addition to existing methods for protease substrate specificity characterization due to its ease of use and the level of detail of specificity information that can be obtained for active enzymes under well-established conditions. However, I also get the impression that it will not be a game-changer. I am missing a convincing demonstration of the posited merits of the method in protease de-orphanization, evidence for successful

identification of the rarely seen substrate cooperativity beyond neighboring sites or truly novel biological insights that would appeal to the broad readership of Nature Communications. A major limitation in de-orphanizing substrates is the difficulty to obtain pure, active enzyme as required for the HTPS assay (see also point 4 below). All proteases investigated by HTPS in this study belong to the most extensively studied proteases overall, and the substrate prediction algorithm even incorporates this prior information in scoring the most likely substrate cleavages. Unfortunately, the physiological relevance of the in vivo substrate prediction is not confirmed for a single new substrate.

In summary, I believe that the manuscript will be better suited for publication in a more specialized journal.

Concerns

1. In HTPS, native lysate is incubated at a high protease:substrate ratio of 1:50 and only peptide products with a MW <10 kD can be detected. Unsurprisingly, for most proteins, multiple peptides are identified, indicating that protein structure may have been compromised after initial cleavage. Did the authors test whether any proteins remain correctly folded under these conditions?

2. Line 320 / Fig 4C: The number of identified cleavages is taken as a measure of activity. If this is true, a varying number of identified peptides should be observed at different incubation times – is this the case? And if yes, does the apparent substrate selectivity reflect kinetic preferences?

3. The apparent switch in P1 substrate specificity from basic K/R to D/W and D/H residues reported for aFVII and aFIX, respectively, are truly remarkable and must be substantiated. Is this true specificity, or could this result from background activity in the sample that becomes apparent with the strong decrease of activity of the test protease due to a lack of sodium ions?

4. No distinction can be made between true cleavages and cleavages pre-existing in the substrate background. As demonstrated in Fig S3, this may not be problematic if the percentage of this preexisting cleavages is low in assays with very active enzymes. However, this will be more problematic when assessing specific enzymes such as TEV and result in false positive identifications that cannot easily be accounted for.

5. The substrate prediction algorithm incorporates prior information (co-citation) in scoring the most likely substrate cleavages, which could result in a bias labeling interacting or merely co-expressed

proteins as more likely substrates. Utility should be proven by demonstrated identification of a physiological relevant substrates.

6. Lines 473-75: The number of new peptide cleavages is vastly exaggerated and mostly irrelevant, as this includes the peptide counts derived from cleavage with proteases commonly used in proteomics. Any large-scale proteomics dataset generated with trypsin would easily surpass the substrate IDs reported in MEROPS!

Minor comments:

- Introduction, line 76 ff. The authors very loosely define degradomics as exposure of test samples to proteases of interest. This definition is too narrow (PMID: 12094217). More importantly, the following sentences can be misunderstood that all previous techniques are also limited to the analysis of such in vitro experiments.

- Please also discuss limitations of HTPS compared to existing methods: Most methods for protein termini profiling by negative selection can be used to analyze tissues with differing protease activity, as for example knock-out and wt mice, and thereby provide insights into in vivo proteolysis, which HTPS cannot.

- Line 122: “The results expand the repertoire of known substrates/cleavage events for coagulation proteases by about two orders of magnitude, from 428 to 38513”. Emphasize that this is limited to the number of cleavage sites listed in MEROPS, which does not include several hundred of cleavage sites that have been reported for human factors FIXa and FXa using PICS but are not included in MEROPS (PMID: 21846260, PMID: 30644641).

- Line 288 “...extend the knowledge on structural determinants of protease specificity” – please describe the new aspects extending the knowledge on structural determinants of thrombin specificity

- Line 616: Please clarify: Were peptides from potential contaminants just not considered for the specificity analysis, or not considered at all during peptide identification?

- Line 618. Please move this section to line 608. How many replicate HTPS analysis/in solution digestions with Trypsin, Lys-C, Asp-N, Glu-C and chymotrypsin were considered during this database assembly?

- Line 691: Please clarify: Did you use all MEROPS substrates, or only those annotated as physiologically relevant?

Point-by-point reply to the reviewers

Reviewer #1 (Remarks to the Author):

This study reports a simple proteomic method to characterize protease specificity using products from native lysates obtained under physiological conditions and analyzed by mass spectrometry. The Authors claim that "the method is significantly faster, cheaper, technically less demanding, easily multiplexed and produces accurate protease fingerprints". Particular attention is devoted to proteases of the blood coagulation cascade for which the Authors obtain "substrate profiles of unprecedented depth". Attention is also paid to characterize "entropy" and "allosteric changes" of the protease and the prediction of physiological substrates.

Overall, I find the study interesting as a methodological tool, but less so as a conceptual advance to the field. My suggestions for revision are largely limited to the coagulation proteases and some interpretations.

1. It is unclear what the Authors mean by "entropy". For sure they do not refer to a thermodynamic quantity but they also provide no measure of this quantity, context and practical meaning.

REPLY: The entropy in context of proteases and protease research refers to the strictness/looseness of protease cleavage preferences, mostly independent of the number of detected protease cleavages as described by Fuchs and coworkers (Fuchs *et al.*, 2013, doi: 10.1371/journal.pcbi.1003007). In this context, the cleavage entropy is calculated as information entropy (Shannon entropy) and not as a thermodynamic value. The cleavage entropy allows ranking of different proteases with respect to their specificity and is thus a measure to classify proteases from unspecific (with high cleavage entropy) to highly specific (with low entropy).

We acknowledge that this may lead to confusion with the thermodynamic entropy, and we therefore replaced the term "entropy" typically associated with thermodynamics with the term "cleavage entropy" throughout the manuscript. Furthermore, we included a better explanation of the meaning of this quantity when it is introduced in the paper on page 8.

2. Blood coagulation proteases, especially the vitamin K-dependent ones, are lousy enzymes without their physiological cofactors. Were cofactors used in the assays? For example, the activity of thrombin toward protein C without thrombomodulin is nearly unmeasurable. Ditto for the intrinsic activity of FVIIa without tissue factor. It is highly problematic to establish entropy and allosteric effects when working with poorly active enzymes. To that end, were enzymes titrated for activity?

REPLY: This is indeed a very important remark, as comparing proteases with confirmed/determined level of activity is an absolute necessity for any protease activity assay/screen. While we did not include this information in the first draft of the manuscript, we always determined the activity of all blood proteases before we used them to study specificity, cleavage entropy and allostery. The activity was determined by active site titration with use of specific serine protease inhibitors like TPCK, PPACK and GGACK and the corresponding fluorescent substrates. We included the protease activity information in the first supplemental table and the active concentrations of proteases determined by active site titrations are now included in the resubmitted version of the paper (Table S1). Importantly, all

coagulation proteases were used at 1/50 $[E_a]/[S]$ ratio in the assays where the active concentration $[E_a]$, determined with active site titration was used to calculate the final concentration of active protease needed for the assay thus ensuring a correct comparison of the proteases.

For cost and time reasons, we could not repeat all HTPS protease screening in presence of the suggested co-factors, but we tested the activity and specificity changes of aFVII (Figure 4A) and α -Thrombin (Figure R1A, please see at the end of the document) in presence and absence of cofactors (Tissue Factor and Thrombomodulin, respectively). In the experiment, we incubated the cofactor and the protease for 30 min at 10 °C using a 10-fold excess of their physiological cofactors, before using the protease for the HTPS screen. While α -Thrombin and aFVII show substantial activity even without the respective co-factors, we observed for aFVII a significant 3-fold increase of detected cleavages in presence of Tissue Factor.

3. The "allosteric" effect of Na^+ needs attention. In the case of thrombin, Na^+ does not change specificity from fibrinogen to protein C (Figure 4A is decades old and potentially misleading), but simply increases the activity toward most substrates. The change in "specificity" from fibrinogen to protein C is produced by thrombomodulin binding and is entirely a steric effect that excludes one substrate and let the other one in. Whether Na^+ binding changes the specificity profile of thrombin remains to be seen, especially in the presence of thrombomodulin. Similar arguments apply to FVIIa, FIXa, FXa and activated protein C in the presence of tissue factor, FVIIIa, FVa and protein S.

REPLY: As the message in the model shown initially could be misleading or outdated to some extent, we removed it. We acknowledge the reviewer for this comment and as suggested, we repeated the entire allostery experiment at 20 °C (please see also answer 5). Under these conditions which are suboptimal for proteolytic activity (Figure S6A, C), we can be sure that a high percentage of α -Thrombin population is bound or not bound with Na^+ in the corresponding binding site. We assayed the allostery effects in experiments with 0.2 M NaCl, 0.2 M ChCl and 0.2 M LiCl. Moreover, in case of α -Thrombin we performed the allostery experiment in presence of NaCl, LiCl and ChCl with and without Thrombomodulin (TM). In the presence of cofactor, we incubated α -Thrombin with 10-fold excess of TM for 30 min at 10 °C. For the sake of clarity we omitted the condition with TM in the main figure (Figure 4B, 4C), but we report the complete experiment in Figure R1B (please see the end of the document). Here, the HTPS screen provided 2 levels of information: i) activity change and ii) specificity change after binding of Na^+ to the allosteric site.

i) Change on the level of activity: In presence of NaCl, α -Thrombin was more active in comparison with the conditions with LiCl, ChCl and ChCl+TM (an average of more than 9000 versus ~6000 cleavage events were observed). This reflects the conformational change of α -Thrombin: when the protein is coordinated with Na^+ (fast form) it is more active than the condition without Na^+ (slow form). Of note, we didn't observe a significant effect on the level of activity after incubation of TM with α -Thrombin in the slow form (Figure R1A).

ii) Change on the level of specificity: To evaluate the change of specificity, we first calculated the amino acid preference per position from all cleavage patterns identified in the experiment. Specifically, we calculated the preference of AA per position as significant fold of change compared to the random distribution of AA in the proteome. In a second step, we calculated for the two important substrates of α -Thrombin, namely FGA (LAEGGVR↓GPRVVERH) and PC (QEDQVDPR↓LIDGKMTR) the sum of significant fold change of enrichment of AA per position (i.e. HTPS Motif Score). HTPS Motif Score provides

the preference of α -Thrombin forms towards one substrate compared to the other. Of note, we found that there is a preference of α -Thrombin in the fast form for FGA compared to PC. This preference is less evident in case of α -Thrombin in the slow form (in this case, both scores have similar values). Importantly, we observed that the TM cofactor didn't change the specificity of α -Thrombin in the slow form (Figure R1B). This could be seen as a limitation of the method, which infers changes of active site (AS) specificity using cleavage sites identified from an "artificial" cell lysate as library of substrates. This condition (far from being *in vivo* extracellular environment) ignores the presence of all physiological interactors (i.e. TM), complex or inhibitors, which can modulate the activity of the protease *in vivo*. This could be the case of TM, which shifts the specificity of α -thrombin for PC with a steric hindrance mechanism.

Altogether, the information for protease activity and specificity confirm the model in which Na^+ modulates the selectivity of α -Thrombin substrates for pro coagulant activity (FGA) and anticoagulant activity (PC) (Dang *et al.*, 1995, doi: 10.1073/pnas.92.13.5977 and De Filippis *et al.*, 2005, doi: 10.1042/BJ20050252). This is most likely the consequence of 2 distinct events (activity and specificity): first, α -Thrombin in the fast form is more active in comparison to the slow form; second, α -Thrombin in the fast form has a clearer specificity-based preference towards FGA versus PC, which is lost in presence of ChCl (slow form).

4. The strategy of characterizing the Na^+ effect from studies that compare NaCl vs ChCl definitely works for thrombin but not in general. It has been known for two decades that FXa is specifically inhibited by ChCl and quaternary amines. Other coagulation proteases may be inhibited by ChCl as well. The "no Na^+ " conditions should be produced by either eliminating Na^+ (no constant ionic strength) or adding LiCl as "inert" salt.

REPLY: We are aware of this potential problem and aFX is indeed inhibited by ChCl, (Rezaie *et al.*, 2000, doi: 10.1021/bi992006a) as the positive quaternary ammonium group of Ch^+ favorably couples to the negatively-charged active site cleft and this can function as a competitive inhibitor. While ChCl can act as an inhibitor in case of aFX, LiCl is too small to produce an allosteric effect in the Na^+ binding site. We conducted an extensive literature search to see if other coagulation proteases included in the study are sensitive to ChCl. Other than α -Thrombin, where it is well known that the Ch^+ ion stabilizes the slow form (Di Cera 2008, doi: 10.1016/j.mam.2008.01.001), we found that for aFIX and aFX decreases the values of specificity constant (k_{cat}/K_M) in comparison to Na^+ condition (Dang *et al.*, 1996, doi: 10.1073/pnas.93.20.10653, Gopalakrishna *et al.*, 2006, doi: 10.1160/TH06-03-0156).

To address these concerns from the reviewer we assayed the activity and specificity changes of aFX, aFIX and α -Thrombin in presence of 0.2 M NaCl, 0.2 M LiCl and 0.2 M ChCl. Remarkably, using ChCl and LiCl as control, we obtained very similar results in terms of activity (Figure 4E) and specificity (the proteases with ChCl and LiCl as control cluster together in the Pearson correlation matrix, Figure 4F). These results highlight the lack of specific interactions in case of ChCl and LiCl with the Na^+ binding site and consequently no allosteric effect.

5. Assays were carried out under physiological conditions, but the pH used was 7.8. Why not 7.4? Also, Na^+ binding to coagulation proteases is weak and very temperature dependent. For some proteases, 200 mM NaCl at 37 C means very little Na^+ bound. If the goal is to establish the effect of Na^+ on specificity, conditions should be used where binding of Na^+ is maximized, increasing NaCl and working at 20-25 C.

REPLY: In the previous batch of experiments, we characterized the activity and specificity of all proteases using conditions commonly used in proteomic experiments (20mM ammonium bicarbonate, pH 7.8 and 37 °C). The temperature of 37 °C mimics the physiological conditions under which proteases operate in humans. Nevertheless, for the allostery experiment, we carefully considered the reviewer's comment regarding the possibility of assaying allostery under conditions that maximize Na⁺ binding. We therefore repeated the experiment according to reviewer's suggestions. Please note that we generated a new dataset for the allostery experiment, and all plots in Figure 4 were generated from this. We performed the experiment at 20 °C to be sure that at 0.2 M NaCl most of α-Thrombin is in fast form. Importantly, before we repeated the allostery experiment, we checked the influence of the temperature on the specificity and activity of α-Thrombin and Chymotrypsin (at 20 °C, we observed that proteolytic activity was reduced by 20%, but we did not detect changes on the level of specificity) (Figure S6A, B, C, D). In a further step, we repeated the allostery experiment at 20 °C for all proteases (i.e. 11) in triplicates and included double controls with LiCl for three proteases to generate a dataset that systematically investigated in an unbiased way the allosteric effects for all coagulation proteases (in total around 90 mass spectrometry runs). From this dataset, we excluded PC as at 20 °C the proteolytic activity of PC was low, resulting in a small number of detected cleavages from which it was difficult to extract the activity and specificity features.

Remarkably, even if it is not possible to make a direct comparison between the old (37°C, pH 7.8) and the new dataset (20°C, pH 7.4), because we used a new batch of lysate, newly ordered batches of proteases, different MS instruments and a modified LC-MS set up, besides a different temperature and pH, we obtained results in line with the old dataset, essentially leading to the same conclusions. In both cases, we correctly identified proteases, which can coordinate a Na⁺, as these show bigger changes of activity and specificity, compared to the control, showing that HTPS is an excellent approach to screen changes of proteases activity and specificity as result of modulators. In fact, it is sensitive enough to even investigate effects of allosteric binders.

6. It should come as no surprise that a protease may have a specificity profile much wider than suspected from its known panoply of physiological targets. Does that mean that other unknown targets of physiological interest may be discovered? And would that be facilitated by known cofactors? The Authors may want to comment on that.

REPLY: We acknowledge this valuable comment. Indeed, many proteomic methods in the past focused on generating specificity profiles of proteases (e.g. PICS, DIPPS) and identified substrates of proteases acting on complex proteomes (e.g. TAILS, COFRADIC). These have shown that the protease specificity profiles and thus the potential pool of substrates is much broader than the display of a true ensemble of physiological substrate cleavages that actually occur *in vivo*. While many past methods struggled to overcome the challenge of sufficiently mimicking physiological conditions, our method works strictly with native protein samples, thus substantially increasing the probability of correctly extrapolating the results from the HTPS screen to more physiological substrate conditions/contexts. For the analysis that is presented in Figure 6, we used the substrate specificity data inferred from identified cleavages in the native cell lysate and applied a series of simple filtering steps to i) correctly identify several known physiological substrates for α-Thrombin (Figure 6C), ii) comprehensively map a large portion of known protease-substrate relationship of the intrinsic and extrinsic coagulation pathway (Figure 6K) and iii) predict new substrate candidates (Table S8).

As an example of a potential substrate candidate, we validated this with a cleavage assay of complement component C3 with α -Thrombin. We investigated the overlap of the top scoring predicted substrate cleavage events and the ones detected in the cleavage assay (Figure 6E, 6F). Indeed, we confirmed 5 out of 11 C3 cleavage sites showing that HTPS data can serve as a valuable resource to facilitate the discovery/guide the selection of potential physiological protease substrates. Remarkable, all cleavage sites for C3 protein (17 in total) were predicted with an HTPS motif Score > 2 (top 5% hits), indicating that HTPS is a good proxy to successfully map protease cleavage products.

C3 has been already reported as substrate of several blood proteases (Amara *et al.*, 2010, doi: 10.4049/jimmunol.0903678) and α -Thrombin-generated complement factor cleavage fragments could activate the complement pathway (Krisinger *et al.*, 2012, doi: 10.1182/blood-2012-02-412080, Huber-Lang *et al.*, 2006, doi: 10.1038/nm1419).

A supplemental table (Table S9) reporting the cleavage events detected with di-methylation strategy for C3 cleavage with α -Thrombin was added to the revised paper. A map showing the overlap of HTPS Motif Score-based cleavage motif prediction and the identified cleavages with a modified TAILS reductive di-methylation protocol was included into Figure 6E.

Reviewer #2 (Remarks to the Author):

The manuscript by Uliana and Vizovisek describes an easy-to-execute, high-throughput screening method to profile protease specificity. Overall, I like this work because of its simplicity and some smart tricks, such as using a search space that is tailored to the used sample. In short, the method is based on treating a cell lysate with proteins in their native conformation with a protease of interest, followed by filtering over a 10 MWCO filter, which isolates the digested peptides, which are then directly analyzed by LC-MS/MS.

At first, the authors benchmark their method against several proteases commonly used for digestion in proteomics, as well as two MMPs. Then, they use different proteases from the blood coagulation cascade to profile specificity and investigate the role of sodium binding as allosteric regulator. Furthermore, they use the found protease specificity information to design selective protease substrates and search for native substrates.

I think that the experimental design is done well, the data analysis is also performed well and conclusions are based on the data. The authors provide scripts online that will help other laboratories to implement this workflow.

Because of the simplicity of the sample preparation, I think the method will be easily transferable to other laboratories and therefore attract broad interest of researchers in the field of proteolysis. I therefore think that Nature Communications is an appropriate journal, and I recommend publication of this manuscript after a minor revision as detailed below:

Comments:

1. P3 line 57, and also P20 line 489: the authors state that HTPS is used in near-native conditions. I don't like this term. What is a near-native condition? In section 2.3 blood coagulation cascade proteases are used on HEK cell lysates. As blood coagulation proteases usually only encounter blood serum, this really cannot be called near-native conditions (the authors state this also in their results section). Furthermore (although I don't want to be too nitpicking), the protein concentration is 1 mg/mL, which is approximately a factor 200 lower than native conditions in HEK cells. HEPES? Not native etc. I think that a better phrasing would be: HTPS is used on lysates with proteins in their native fold (and I guess that is what the authors mean). But I anyway think that this term should not be emphasized too much, because I doubt if it really matters for the results. The strength of the study is: an amazingly simple method for protease substrate specificity profiling.

Whether the substrate proteins are present in their native fold or in a different conformation, I don't expect that this would have a big influence on the data.

REPLY: We thank the reviewer for this insightful comment. We adapted the wording accordingly and emphasized that the proteins in the assay are present in their native fold, a crucial prerequisite to investigate the protease-substrate relationships in a setting that can potentially mimic near-native conditions.

The reviewer also suggested that it is not expected that the native fold or other conditions would have a big influence on the data. This remark is definitely true for determination of protease specificity. Accordingly, we observed this when comparing protease specificity profiles obtained for MMP-2 and MMP-3 with HTPS with profiles obtained with other methods including PICS, TAILS and DIPPS. While these methods have substantially different workflows starting from a peptide library (PICS), a complex proteome sample (TAILS), a gel-

separated denatured proteome (DIPPS) or a whole proteome sample preserving proteins in their native fold (HTPS), they all yielded a remarkably similar specificity patterns for MMP-2 and MMP-3.

On the other hand, previous work indicates that this may not hold true for other applications. This was suggested on several occasions even in the pioneering proteolysis experiments by Cleveland and coworkers (Cleveland *et al.*, 1977, J. Biol. Chem. 252, 1102-110.) where proteolytic cleavage patterns were for the first time used as a readout. Later, the proteolysis studies of BSA with Chymotrypsin in presence/absence of low concentrations of SDS demonstrated that the folded state of protein substrate has a dramatic effect on the proteolysis pattern observed (Walker and Anderson, 1985, doi: 10.1016/0003-2697(85)90403-8). While these experiments were conducted several decades ago, they clearly emphasize the importance of studying protease substrate cleavage patterns on proteins in their native fold. This was further substantiated with the development of limited proteolysis workflows where native proteins are exposed to a high concentration of protease in a relatively short time to reveal the structural features and compare the changes of these protein structural features over a series of near-native conditions (Schopper *et al.*, 2017, doi: 10.1038/nprot.2017.100). Because our method uses native cell lysates preserving native protein structures, it has significant potential to extrapolate the findings to physiological systems. To underline the importance of this, we used HTPS to study allostery changes of proteases in presence of Na⁺.

2. P5, line 89: note that ChaFraTip can also be used on non-digested proteins/cellular samples (see: Shema, Zahedi *et al*, Mol Cell Proteomics 2018)

REPLY: We agree with this remark and corrected it in the text accordingly.

3. P7, line 145: “which retains undigested proteins and the added protease and supports recovery of the cleavage products in the flow-through.” This assumes that all proteolytic cleavages will lead to cleavage products that are lower than 10 kDa. I agree that this will indeed be the case with the test proteases trypsin, GluC, etc. But I am convinced that this is not the case for all proteases and all substrates: Sometimes a very small part (propeptide) is cut from a protein and it virtually retains its original MW. Or in case of very limited proteolysis, fragments would be larger than 10 kDa. In the discussion (line 512) the authors speculate about viral proteases, but these usually cut one large viral protein into large fragments (> 10 kDa) that form functional proteins. I think the authors should briefly mention/discuss this in their introduction or discussion, and also remove their statement about viral proteases in the discussion, because I don't think this would be an appropriate application (especially if you would work on with the native viral protein). On a human cell lysate (folded or denatured), it may lead to sufficient cut sites to generate small enough peptides to be isolated in the flow through.

REPLY: The filter device is used as a step to increase simplicity and speed of the protocol and it could indeed represent a limitation in case of proteases with a very specific activity or proteases that generate fragments that would be bigger than 10 kDa and would thus not pass the filter. In several different studies with various different proteases (ranging from standard proteases used in proteomic workflows to less studied/used proteases) it was shown, however, that these usually generate an extensive repertoire of fragments that are amenable for MS analysis (Giansanti *et al.*, 2016, doi: 10.1038/nprot.2016.057, Vidmar *et al.*,

2017, doi: 10.15252/embj.201796750, Soh *et al.*, 2020, doi: 10.1021/acs.analchem.9b03604).

Nevertheless, to address this potential limitation, we tested an additional experimental and data analysis workflow to characterize proteases that would potentially generate fragments too big to be recovered and thus analyzed with standard HTPS (Figure R2A, B). In this case, the simplicity of the setup enables a sequential use of two proteases (the “unknown” protease to be characterized marked as “RED PROTEASE”, and a second protease with a well-characterized protease specificity, i.e. Trypsin). To demonstrate the validity of this setup, we applied it to AspN as a case study, as the protease has a very high specificity, which is complementary to Trypsin. In the experimental pipeline we added to the pre-existing control (control matrix, a matrix generated from the random distribution in the proteome) an additional matrix which includes cleavages generated by Trypsin. The differential analysis of frequency of AA per position between the protease matrix and the two control matrixes (control matrix, trypsin control) generates the new protease fingerprint. The scatter plot shows the specificity, which is obtained as result of such experiment and can be used to identify the specificity of the “unknown” protease. We of course cannot completely exclude that in case of proteases that are non-frequent cutters the assay would require some further optimization to find optimal conditions for a double digestion with a well-known and a non-characterized protease. Furthermore, it could also require parallel use of more than one different complementary proteases to retrieve an accurate specificity of a non-frequent cutter. Nevertheless, this experiment indicates that the potential cut-off limitation of our setup can be, in general, overcome by a simple adjustment of the protocol. While it might be the case that for some very specific/atypical proteases the assay might not be the method of choice, we performed additional experiments to further demonstrate the applicability of HTPS also for proteases with high specificity, that are not expected to be frequent cutters (and could possibly generate larger fragments) and, as consequence, compromise this analysis strategy.

To demonstrate this, we included NS3 West Nile virus protease to the screen (Figure 2B). This experiment demonstrated that HTPS performed well even with a viral protease that is not expected to be a very frequent cutter and enabled us to determine its cleavage specificity and cleavage entropy in a rapid way. Due to extremely limited amount of the protease available, we perform only one single digestion. Nevertheless, the results obtained with HTPS were in good agreement with a recent study that utilized a hybrid combinatorial substrate library to probe the positional preference of the NS3 West Nile virus protease (Rut *et al.*, 2020, doi.org/10.1016/j.antiviral.2020.104731). Importantly, both methods reported that WN NS3 has a trypsin-like specificity in position P1, with extended substrate preference for Lys at position P2 and P3. While hybrid combinatorial substrate libraries can include non-natural amino acids in the screen which is beneficial especially when designing selective activity-based probes, this technique is limited mostly to specialized laboratories that have access to the comprehensive substrate libraries required to conduct such profiling. Here, in terms of general applicability for protease research, HTPS has a clear advantage, since it allows the generation of a reliable specificity profile with minimal amount of proteases in a simple and straightforward way.

4. Block cleavage entropies -> I have no idea what this really means, and I think that it will have no meaning to a reader, if it is not explained in the figure caption of Figure S6. I briefly took a look at Reference 35, and as far as I understand, these numbers reflect if the protease relies on the occurrence of specific successive amino acids in small blocks along the cleavage site. I think the authors should mention this and also indicate what the numbers on the y-axis mean (i.e. what is their implication if they are higher or lower). There also seems to

be a mistake in the figure caption of S6, because the block entropy values are not given per position (as written in the caption), but per block (as indicated in the graphs). Please indicate what B4, B3, B2 etc stands for.

REPLY: We apologize for not explaining this more clearly. We realized that this terminology is not widely known outside the immediate protease community and we tried to define explicitly its meaning. A block-based entropy calculation for protease-generated substrate peptides can reveal the successive combinations of amino acids that are particularly favored by the investigated protease. The block entropy analysis is used to discover prominent blocks (i.e. amino acid motifs that are particularly enriched in a cleavage dataset), thereby evaluating the extent of sub-site cooperativity in the detected cleavage motifs. The block nomenclature refers to sub-site positions included in a block (amino acid stretches) where B1 includes position P1, B2 positions P2 and P1, B3 positions P3, P2 and P1 and so forth, while block B1' includes position P1', block B2' includes position P1' and P2', block B3' includes positions P1', P2' and P3' and so forth. We included a better explanation of the block entropy in the paper.

The figure caption of the Supplemental figure S6 (now Figure S10) has been corrected accordingly.

5. P16, line 396: S2238 does not only have a non-natural P3 D-Phe, but also a P2 Pip (“homoproline”) as non-natural amino acid, likely affecting the selectivity. Please mention.

REPLY: We thank the reviewer for pointing this out as it was indeed not mentioned in the paper. We corrected in the text accordingly and highlighted the change in red color for better tracking.

Reviewer #3 (Remarks to the Author):

The authors present a method for high-throughput profiling of protease cleavage site specificity based on a standard in solution digest of isolated, non-denatured proteomes under semi-native conditions, essentially substituting trypsin by the test protease of interest in filtered-aided sample preparation. In “High-Throughput Protease Screen” (HTPS), cleavage products are isolated by spinning through a 10 kDa filter, followed by mass spectrometry-based identification. Unspecific peptide sequence matching was constrained using a database limited to proteins identified by the same method using standard digestion enzymes. This resulted in the identification of several hundred to thousands of cleavage sites for each tested protease. These rich datasets were used for an in-depth characterization of cleavage site specificity using a combination of previously established data analysis routines, including analysis of preferred substrates using iceLogos, calculation of cleavage entropy and block entropy as comparative measures for specificity and cooperativity of neighboring sites.

The serine proteases of the blood coagulation system were chosen for proof of concept analysis, and the results obtained using HTPS recapitulated prominent features documented in the literature. Based on the HTPS results, two fluorescently labeled tetrapeptide substrates were designed to successfully exploit a differences in P2 subsite selectivity between aFX and thrombin, respectively. Finally, candidate physiological substrates were predicted based on the match in specificity, followed by a scoring scheme that utilizes literature knowledge such as predicted co-localization, co-expression, and co-citations.

HTPS appears technically sound and will be a useful addition to existing methods for protease substrate specificity characterization due to its ease of use and the level of detail of specificity information that can be obtained for active enzymes under well-established conditions. However, I also get the impression that it will not be a game-changer. I am missing a convincing demonstration of the posited merits of the method in protease de-orphanization, evidence for successful identification of the rarely seen substrate cooperativity beyond neighboring sites or truly novel biological insights that would appeal to the broad readership of Nature Communications. A major limitation in de-orphanizing substrates is the difficulty to obtain pure, active enzyme as required for the HTPS assay (see also point 4 below). All proteases investigated by HTPS in this study belong to the most extensively studied proteases overall, and the substrate prediction algorithm even incorporates this prior information in scoring the most likely substrate cleavages. Unfortunately, the physiological relevance of the *in vivo* substrate prediction is not confirmed for a single new substrate.

In summary, I believe that the manuscript will be better suited for publication in a more specialized journal.

REPLY: We appreciate the comments of the reviewer but respectfully disagree with his/her assessment of the significance of the work. We believe that the method constitutes a significant advance in the field, for the following reasons: First, the simplicity and robustness of the method enables a high-throughput parallel testing of many proteases/conditions/co-factors with a low amount of protease required (e.g. testing the protease dependency on co-factors), as we perhaps best demonstrated by the extensive allostery screen. Second, the rich protease cleavage data generated on proteins in their native fold allow the data-driven reconstruction of protease activity, specificity and cleavage entropy features and thus enable sensitive comparison of proteases as exemplified by the analysis of α -Thrombin proteoforms

(γ -Thrombin differs from α -Thrombin for only 3 amino acids). Third, HTPS data can serve as a proxy to predict potential protease substrates for validation in biochemical assays and to select substrate candidates in orthogonal systems as exemplified by blood coagulation proteases.

We are convinced that HTPS workflow *per se* represents a major advancement in the field, as similar strategies could be readily applied/adopted to other proteases/proteolytic systems to aid protease substrate discovery and thus advance systems biology research of proteases.

Concerns

1. In HTPS, native lysate is incubated at a high protease:substrate ratio of 1:50 and only peptide products with a MW <10 kD can be detected. Unsurprisingly, for most proteins, multiple peptides are identified, indicating that protein structure may have been compromised after initial cleavage. Did the authors test whether any proteins remain correctly folded under these conditions?

REPLY: It is expected that after the initial cleavage event the proteins will often partially unfold thereby exposing and making more accessible other potential cleavage sites that would be otherwise buried and thus not accessible for proteolysis. This is also how physiological proteolysis functions in living cells and proteins that have been already cleaved by a protease are more prone for further cleavage events or are marked for degradation by the ubiquitin-proteasome system. Nevertheless, we suspect that the effect of protease cleavages (particularly in case of more specific proteases) is not sufficient to unfold proteins and protein fragments completely as is the case in presence of denaturants (Urea, SDS).

To investigate this, we monitored the loss of fluorescence signal of GFP as an indicator of the protein folding; we monitored the fluorescence signal of rGFP (2 μ g) in a mixture of 5 proteins (Conalbumin, Thyroglobulin, Ovalbumin, Human serum Albumin and Aldolase). We monitored in triplicates the fluorescence intensity of rGFP over 4 hours in 3 minutes intervals and in different conditions: native conditions with and without protease (α -Thrombin at a 1:50 [E]/[S] ratio) and in denaturant condition (heat denaturation 90 °C, 8 M Urea) (Figure R3). We observed that in presence of proteases, the intensity of fluorescence is not substantially reduced in comparison to the intensity drop seen in the denaturing conditions. Importantly, in native conditions the slope of the fluorescence intensity decreasing over time was not significantly different between native condition without and native condition with the protease. This suggests the effect of fluorescence quenching is likely higher than the effect of fluorescence signal loss as result of protein unfolding caused by proteolytic cleavages. While GFP is a very stable protein, we can assume that under the conditions used in the assay the presence of protease did not lead to substantial denaturation/unfolding even after 4 hours of proteolysis.

2. Line 320 / Fig 4C: The number of identified cleavages is taken as a measure of activity. If this is true, a varying number of identified peptides should be observed at different incubation times – is this the case? And if yes, does the apparent substrate selectivity reflect kinetic preferences?

REPLY: We thank the reviewer for this observation, which enabled us to further explore the potential and features of HTPS. To address this point, we performed an HTPS experiment

with α -Thrombin and Chymotrypsin and monitored the number of detected cleavages as a function of the digestion time (Figure S5A, C). Indeed, we detected that the number of cleavages increased with time thus demonstrating that this readout can be used to evaluate protease activity. While the numbers of detected cleavages identified with HTPS were proportional to the activity of the protease (incubation time), the specificity inferred from HTPS data at different time points did not change significantly over time (Figure S5B, D). This supports the idea that when comparing proteases at same active concentrations and digestion times and especially when comparing the same protease under different conditions we can draw conclusions regarding their activity as exemplified by the case of FVII and FVII plus its respective cofactor (Tissue Factor) (Figure 4A).

3. The apparent switch in P1 substrate specificity from basic K/R to D/W and D/H residues reported for aFVII and aFIX, respectively, are truly remarkable and must be substantiated. Is this true specificity, or could this result from background activity in the sample that becomes apparent with the strong decrease of activity of the test protease due to a lack of sodium ions?

REPLY: We performed a thorough evaluation of the peptide background to address this issue. The differential heatmap reported in the supplementary figure (Figure S11) and Figure 4H shows the different preference of AA specificity for proteases in presence and in absence of Na^+ . It is important to keep in mind that this is not an absolute value (like in Figure S7) but a ratio which indicates the preference in both conditions. The ratio used in the protease specificity comparison is calculated from significant (p -value < 0.01) enrichment of amino acid per position compared to the random distribution in presence of NaCl and ChCl. As example, FIX and α -Thrombin keep the preference for K/R in P1 position (Figure S7) but if we compare the specificity, we can observe that while Na^+ form prefers K in position P1, the Na^+ free form prefers R in position P1. The change of specificity could be influenced by a reduced number of cleavages in particular in case of a protease almost not active (FVII in presence of ChCl) but, we excluded that the preference could be generated from the background peptides. We performed the HTPS workflow for the analysis of background peptides (~300 cleavages) and we didn't obtain any specific fingerprint as we did not detect a significant enrichment compared to the random distribution ($-\log_{10}p$ -value > 0.01) (Figure S4B).

4. No distinction can be made between true cleavages and cleavages pre-existing in the substrate background. As demonstrated in Fig S3, this may not be problematic if the percentage of this preexisting cleavages is low in assays with very active enzymes. However, this will be more problematic when assessing specific enzymes such as TEV and result in false positive identifications that cannot easily be accounted for.

REPLY: We agree that in very specific cases this could be a limitation and potentially preclude the robust determination of specificity of a very limited number of proteases. In the case of TEV protease, however, the cleavage motif is very specific, since it cleaves Glu-Asn-Leu-Tyr-Phe-Gln-(Gly/Ser) between the Gln and Gly/Ser. Importantly, this particular cleavage motif is not present in any protein expressed in human proteome (Doerr *et al.*, 2010, doi: 10.1038/nmeth1010-786) and TEV protease is commonly used in protein engineering for specific protein tag removal (because there is no interference with other human proteins). Therefore, profiling TEV protease with human proteome is not likely to yield informative results.

For proteases with very specific cleavage motifs present in human proteome, this problem could be bypassed using a double digestion workflow where a well-characterized protease is used in a complementary way to ensure sufficient extent of proteolytic products for MS analysis (Figure R2).

To further substantiate the level of impact of the proteolytic cleavages in the background, we analyzed the ~300 cleavages identified in the flow through of the HTPS workflow. As now included in Figure S4B, (see also answer 3) the background doesn't have an impact on the determined specificity.

5. The substrate prediction algorithm incorporates prior information (co-citation) in scoring the most likely substrate cleavages, which could result in a bias labeling interacting or merely co-expressed proteins as more likely substrates. Utility should be proven by demonstrated identification of a physiological relevant substrates.

REPLY: The positive value of using prior information on potential protein-protein relationships has been extensively demonstrated in several computational tools. Perhaps the best example is the prediction of kinase substrates as exemplified by NetworKIN (Linding *et al.*, 2008, doi: 10.1093/nar/gkm902). In this case, the software algorithm to predict kinase substrates incorporates data for the phosphosite motif and information of protein proximity using a network generated from STRING database (Szklarczyk *et al.*, 2019, doi: 10.1093/nar/gky1131) (which contains multiple features like text mining, available experimental data, co-expression, neighborhood, gene fusion, co-occurrence and database of protein interactions). Similar to the approach used by NetworKIN, starting from the cleavage site motif, we applied a series of filtering steps to reduce the number of potential substrates using criteria like protein abundance, co-citation and number of possible cleavage per protein for our investigation of protease-substrate relationships.

To further validate the applicability of our filtering approach, we selected complement factor C3 and performed proteolysis with α -Thrombin followed by subsequent mapping of cleavage events utilizing reductive di-methylation strategy used for labeling free N-termini in TAILS protocol (Kleifeld *et al.*, 2008, doi: 10.1038/nprot.2011.382). Remarkably, this experiment yielded a match for 5 out of 11 top scoring HTPS predicted cleavage sites, demonstrating that C3 is a substrate of α -Thrombin. Most importantly, C3 has been already reported as substrate of several blood proteases (Amara *et al.*, 2010, doi: 10.4049/jimmunol.0903678) and studies were performed to get a better understanding of the relationships between blood coagulation and complement system: α -Thrombin-generated alternative complement factor cleavage fragments could activate the complement pathway (Krisinger *et al.*, 2012, doi: 10.1182/blood-2012-02-412080, Huber-Lang *et al.*, 2006, doi: 10.1038/nm1419). Collectively, these data strongly support the correctness of HTPS Motif Score-based prediction of C3 as α -Thrombin substrate.

A supplemental table (Table S9) reporting the cleavage events detected with di-methylation strategy for C3 cleavage with α -Thrombin was added to the revised paper. A map showing the overlap of HTPS Motif Score-based cleavage motif prediction and the identified cleavages with a modified TAILS reductive di-methylation protocol was included into Figure 6E.

In order to test whether we could eventually completely circumvent the need for prior information we tried to predict putative α -Thrombin substrates with a deep learning approach. To train the model we used the cleavage sites identified after α -Thrombin proteolysis (positive set) and a random distribution of cleavages random generated from the HTPS database (negative set). This resulted in 17169 unique sequences. Of these, we employed

3434 to test model performance, 9202 to train the model and reserved 4533 as validation set. The octameric sequences were encoded to numerical integers and then embedded into a 22x128 dimensional feature space. The resulting sequences were then passed through a bidirectional long-short memory term (LSTM) layer with 6 neurons. To mitigate overfitting, L2 regulation (0.001) was applied. Following back and forward passage through the LSTM layer, dropout (0.3) was applied before feeding the output to a fully connected layer with 8 neurons. A single output neuron was then used to predict the probability of an octamer being a substrate of α -Thrombin. The network was trained using the Adam optimizer with a learning rate of 0.001 and binary cross entropy as loss function. The model was build using Keras v2.2.4 (<https://keras.io>) and Tensorflow v2.3.1 (<https://www.tensorflow.org>) as backend in Python 3.6. Loss and accuracy were monitored for every epoch and the model achieving lowest validation loss was selected as the final model.

In Figure R3A, the accuracy and loss per epoch in the training and test set are reported. In the validation set (i.e. not used to train or test the model), the model achieved outstanding accuracy (0.93), recall (0.96) as shown in Figure R3B and an overall ROC AUC of 0.97 (Figure R3C) suggesting that, within the feature space employed the model learned correctly the relationship between peptide sequence and potential α -Thrombin cleavage site motif. However, in the test set the LSTM model predicted 2758 putative substrates (at 1% FPR) from the 280771 sequences derived from the secretome and recovered only a modest number of α -Thrombin substrates reported in MEROPS (9/41). We compared the results obtained from our LSTM model and our HTPS Motif Score: ROC curve shows that HTPS Motif Score outperformed the DL approach (AUC DL=0.65, AUC HTPS Motif Score 0.97) (Figure R3D). With a cutoff of 1% FPR in both conditions, we have an overlap of 13% with the identification of 8 true positive over 41 (Figure R3E); the significant enrichment of true positives indicates the possibility to apply both approaches for the identification of new candidate physiological substrates (Figure R3F).

6. Lines 473-75: The number of new peptide cleavages is vastly exaggerated and mostly irrelevant, as this includes the peptide counts derived from cleavage with proteases commonly used in proteomics. Any large-scale proteomics dataset generated with trypsin would easily surpass the substrate IDs reported in MEROPS!

REPLY: This is a valid point and we corrected this by emphasizing the expansion of cleavage datasets only for coagulation proteases, excluding the counts for proteases commonly used in proteomic workflows. We corrected the text of the discussion accordingly.

Minor comments:

- Introduction, line 76 ff. The authors very loosely define degradomics as exposure of test samples to proteases of interest. This definition is too narrow (PMID: 12094217). More importantly, the following sentences can be misunderstood that all previous techniques are also limited to the analysis of such in vitro experiments.

REPLY: We have corrected this to focus specifically on protease degradomics and emphasized that these techniques have been applied in the past to profile proteolytic events.

- Please also discuss limitations of HTPS compared to existing methods: Most methods for protein termini profiling by negative selection can be used to analyze tissues with differing protease activity, as for example knock-out and wt mice, and thereby provide insights into in vivo proteolysis, which HTPS cannot.

REPLY: Even if HTPS cannot offer a direct insight into *in vivo* proteolysis, the large number of cleavages generated in this study enable the generation of deep protease specificity profiles. This *per se* more than compensates for this potential drawback, as it is possible to combine specificity data generated with a simple filter approach to identify new physiological substrate candidates for validation in biochemical/physiological/*in vivo* assays. This approach can be particularly useful in case of matrices for which it is not possible/easy to perform biochemical assays as in the case of blood plasma (dynamic range abundance, sample amount and accessibility). This was now commented in the discussion.

The extraction of information from databases will be important particularly to investigate proteases in different matrices. This is what we demonstrated in the Figure 6: we start from cleavages obtained in HEK293 cell lysate to determine the specificity of the protease and predict physiological substrates in extracellular region using a series of filtering steps. In a further step, we validated some of the predicted cleavages using an *in vitro* cleavage assay (Figure 6E). We believe that this *per se* is a very important and remarkable contribution as it has been so far very challenging to select a manageable group of substrate candidates for thorough validation. To further demonstrate this and provide an example, we performed a comprehensive analysis of C3 complement factor cleavage products obtained with α -Thrombin to show that several cleavages predicted with our HTPS pipeline could be confirmed in the cleavage assay. Of course, a thorough validation has to be conducted in appropriate cell and animal models where a direct link between the protease and the corresponding substrate can be established either by use of KO or selective inhibition to understand what is the biological significance of detected cleavage events.

- Line 122: “The results expand the repertoire of known substrates/cleavage events for coagulation proteases by about two orders of magnitude, from 428 to 38513”. Emphasize that this is limited to the number of cleavage sites listed in MEROPS, which does not include several hundred of cleavage sites that have been reported for human factors FIXa and FXa using PICS but are not included in MEROPS (PMID: 21846260, PMID: 30644641).

REPLY: This is a valid and valuable comment. Indeed, PICS was used for extensive characterization of FIXa and FXa. Unfortunately, these protease characterization studies are currently not included in MEROPS and therefore this data could not be included in our comparison. We mentioned this as suggested by the reviewer, but we also included the two suggested references when we discuss the specificity profiles of the blood coagulation proteases to further substantiate the correctness of the specificity profiles determined with our method. We highlighted the changes undertaken in red for easier tracking.

One should certainly encourage the community of scientist which work in the protease field to upload their dataset in the database as these could be extremely useful as true positive for the development of better and more accurate ML approaches to identify new possible protease substrates.

- Line 288 “...extend the knowledge on structural determinants of protease specificity” – please describe the new aspects extending the knowledge on structural determinants of thrombin specificity

REPLY: We apologize if this was written in a way that could be misunderstood. What we meant with this was that HTPS can measure cleavage products which reflect structural/conformational changes induced in α -Thrombin and other blood proteases as a consequence of modulators. The section 2.4 was completely rewritten.

- Line 616: Please clarify: Were peptides from potential contaminants just not considered for the specificity analysis, or not considered at all during peptide identification?

REPLY: Our analysis did not include contaminants in the specificity analysis, but they were included in the database searches as per default MaxQuant settings. A short sentence was added in the method section for better clarification (highlighted in the manuscript in red for better tracking).

- Line 618. Please move this section to line 608. How many replicate HTPS analysis/in solution digestions with Trypsin, Lys-C, Asp-N, Glu-C and chymotrypsin were considered during this database assembly?

REPLY: We clarified this issue in the text accordingly. The HTPS screen always included 3 replicates for each protease used in the experiments for constructing the HTPS database (exception for WN NS3, where the amount of available protease was the limiting factor). We also highlighted the changes in text in red for better tracking.

- Line 691: Please clarify: Did you use all MEROPS substrates, or only those annotated as physiologically relevant?

REPLY: Our analysis included all protein substrates for which cleavage events have been reported with different methods including substrates regardless whether they were denoted as physiologically relevant or not. This is mostly for the reason that there is a very limited number of physiological substrates available from MEROPS, even for α -Thrombin (41). Importantly, we excluded synthetic substrates from our analysis.

Figures for the reply to reviewers

Figure R1. The effect of Thrombomodulin on α -Thrombin activity.

(A) The number of detected cleavages for α -Thrombin in presence of NaCl, ChCl or ChCl+TM. (B) The HTPS motif score calculated from HTPS data for substrates of α -Thrombin FGA (LAEGGVRL↓GPRVVERH) and PC (QEDQVDPR↓LIDGKMTR) as the sum of significant fold change of enrichment of AA per position.

«Mapping specificity, cleavage entropy, allosteric changes and substrates in blood proteases by a high-throughput protease screen»

Figure R2. The double digestion HTPS workflow for characterization of proteases.

(A) HTPS workflow enables a sequential use of two proteases: the “unknown” protease to be characterized marked as “RED PROTEASE” and a second protease with a well-known specificity. (B) Application of the workflow to AspN. In this case, the experimental pipeline uses the pre-existing control (control matrix, generated from the random distribution) and an additional matrix, which includes the cleavages generated by Trypsin. The differential analysis of frequency of AA per position between the protease matrix and the two control matrixes (control and trypsin) generates the new protease fingerprint and this result can be used to infer the specificity of the “unknown” protease.

Figure R3. Monitoring GFP fluorescence intensity over different assay conditions.

The GFP fluorescence serves as the indicator of the protein folding state. Denatured GFP completely loses its fluorescent properties, while native GFP produces strong fluorescent signals.

Figure R4. Prediction of putative α -Thrombin substrates with deep learning.

(A) Accuracy and loss per epoch for the training (blue) and validation set (yellow). (B) ROC curve depicting True positive vs. False positive rate. (C) Evaluation of Accuracy, Precision, Recall, F1 score and AUC. (D) Performance comparison of ML results and the HTPS Motif Score. (E) Overlap of predicted substrates generated with HTPS Motif Score and with ML at a cutoff of 1%. (F) p -value of the significant enrichment of true positive targets for HTPS Motif Score and for the ML calculated by a hypergeometric test.

The datasets generated in the course of this study are deposited to ProteomeXchange via the PRIDE database. They can be accessed with the log-in details provided below.

Protease profiling dataset

Project accession: PXD018976

Username: reviewer76711@ebi.ac.uk

Password: RPW4KF6y

PRM measurements of substrate peptides

Project accession: PXD020320

Username: reviewer00479@ebi.ac.uk

Password: tyMybhQK

Allostery dataset (HTPS screen of coagulation proteases at 20°C with NaCl, ChCl, LiCl)
(revision dataset)

Project accession: PXD022959

Username: reviewer_pxd022959@ebi.ac.uk

Password: pwwttlr6

Identification of C3 cleavage sites by α -Thrombin (revision dataset)

Project accession: PXD022971

Username: reviewer_pxd022971@ebi.ac.uk

Password: XSfBSgpk

Characterization of WN NS3 and AspN proteases (revision dataset)

Project accession: PXD022973

Username: reviewer_pxd022973@ebi.ac.uk

Password: pBDVtYCI

Multiple experiments: Analysis of the HTPS FT; Kinetics of Thrombin and Chymotrypsin; Effect of temperature on Thrombin and Chymotrypsin (revision dataset)

Project accession: PXD022972

Username: reviewer_pxd022972@ebi.ac.uk

Password: tRwvmmPL

REVIEWERS' COMMENTS

Reviewer #1 (Remarks to the Author):

All relevant issues have been addressed.

Reviewer #2 (Remarks to the Author):

With their revisions, the authors have sufficiently addressed my remarks and I recommend publication.

Reviewer #3 (Remarks to the Author):

The authors have very adequately addressed my concerns, questions and comments in their extensive and detailed revision. An impressive additional amount of work has been added to demonstrate the capability of HTPS to profile altered activity and allostery-induced changes in high throughput. New data on the West Nile virus protease NS3 and the “double digestion” workflow presented for review only (Figure R2) alleviate my concern that the method may be limited to “frequent cutters”, e.g. proteases able to release sufficiently small peptides by multiple cleavages in the substrate, which may have hampered the broader utility for de-orphanizing uncharacterized enzymes.

A particular strength of the manuscript is the bioinformatic approach to predict potential physiological substrates from cleavage site data using multiple additional sources of information. While I am still missing a demonstration of the physiological relevance of predicted novel substrate cleavages *in vivo*, I understand that such work may be considered to be beyond the scope of this manuscript given the systematic demonstration that the approach recovers known physiological substrates of the blood coagulation proteases with high fidelity.

Overall, I now support publication in Nature Communications.

Some minor comments that the authors may wish to address in their final edits:

- The demonstration that HTPS in combination with double digestion works (Figure R2) is very useful. I would encourage the authors to emphasize this possibility further in the discussion (beyond the current indication in line 162) and indicate if the scripts provided online allow other laboratories to implement this strategy.

- P21, line 523: The assay indeed captures the protease-substrate relationships in the blood coagulation cascade, but while the data preserves directionality of proteases cleavage of other proteases, in my opinion it does not “rebuild the intrinsic and extrinsic coagulation pathway activation” as no functional consequences can be predicted from HTPS cleavage data. Please moderate the language.

- Inconsistent numbers: Line 539 p22 “ten coagulation proteases”, in contrast to other counts as 9 proteases (lines 127, 552)

- I very much appreciated that the authors evaluated prediction of thrombin substrates by deep learning approach (Figure R3, for review only). This nicely demonstrates that including prior information as implemented in the data analysis pipeline presented here is currently still superior, and may be briefly mentioned as a discussion point.

Point-by-point reply to the reviewers

REVIEWERS' COMMENTS

Reviewer #1 (Remarks to the Author):

All relevant issues have been addressed.

Reviewer #2 (Remarks to the Author):

With their revisions, the authors have sufficiently addressed my remarks and I recommend publication.

Reviewer #3 (Remarks to the Author):

The authors have very adequately addressed my concerns, questions and comments in their extensive and detailed revision. An impressive additional amount of work has been added to demonstrate the capability of HTPS to profile altered activity and allostery-induced changes in high throughput. New data on the West Nile virus protease NS3 and the “double digestion” workflow presented for review only (Figure R2) alleviate my concern that the method may be limited to “frequent cutters”, e.g. proteases able to release sufficiently small peptides by multiple cleavages in the substrate, which may have hampered the broader utility for de-orphanizing uncharacterized enzymes.

A particular strength of the manuscript is the bioinformatic approach to predict potential physiological substrates from cleavage site data using multiple additional sources of information. While I am still missing a demonstration of the physiological relevance of predicted novel substrate cleavages in vivo, I understand that such work may be considered to be beyond the scope of this manuscript given the systematic demonstration that the approach recovers known physiological substrates of the blood coagulation proteases with high fidelity.

Overall, I now support publication in Nature Communications.

Some minor comments that the authors may wish to address in their final edits:

- The demonstration that HTPS in combination with double digestion works (Figure R2) is very useful. I would encourage the authors to emphasize this possibility further in the discussion (beyond the current indication in line 162) and indicate if the scripts provided online allow other laboratories to implement this strategy.

REPLY: We thank the reviewer for his comment. We now mentioned in the discussion that our protocol is also suited to incorporate parallel or sequential digestion steps, which might be beneficial in case where the protocol was to be applied for proteases that generate low number of cleavages.

- P21, line 523: The assay indeed captures the protease-substrate relationships in the blood coagulation cascade, but while the data preserves directionality of proteases cleavage of other proteases, in my opinion it does not “rebuild the intrinsic and extrinsic coagulation pathway activation” as no functional consequences can be predicted from HTPS cleavage data. Please moderate the language.

REPLY: We moderated the claim as suggested and removed it from the discussion.

- Inconsistent numbers: Line 539 p22 “ten coagulation proteases”, in contrast to other counts as 9 proteases (lines 127, 552)

REPLY: We thank the reviewer for pointing out this inconsistency that we now have corrected accordingly.

- I very much appreciated that the authors evaluated prediction of thrombin substrates by deep learning approach (Figure R3, for review only). This nicely demonstrates that including prior information as implemented in the data analysis pipeline presented here is currently still superior, and may be briefly mentioned as a discussion point.

REPLY: We appreciate this comment; The concept of using prior information and its importance for predicting protease substrates have been extensively discussed in the results and in the discussion section outlining also that in the future they might be beneficial to aid the development of machine learning approaches for protease substrate prediction/research. Nevertheless, we believe that protease substrate prediction analysis with a machine learning approach should be further implemented and would go beyond of the current scope of the paper.